# Applying time series analyses on continuous accelerometry data—A clinical example in older adults with and without cognitive impairment

**Torsten Rackoll**[1,2,3☯]*, **Konrad Neumann**[4,5☯], **Sven Passmann**[1], **Ulrike Grittner**[4,5], **Nadine Külzow**[1,6], **Julia Ladenbauer**[7], **Agnes Flöel**[2,7,8]

**1** Charité–Universitätsmedizin Berlin, Corporate Member of Freie Universität Berlin, Humboldt–Universität zu Berlin, and Berlin Institute of Health, NeuroCure Excellence Cluster (NCRC), Berlin, Germany, **2** Charité–Universitätsmedizin Berlin, Corporate Member of Freie Universität Berlin, Humboldt–Universität zu Berlin, and Berlin Institute of Health, Center for Stroke Research Berlin (CSB), Berlin, Germany, **3** Charité–Universitätsmedizin Berlin, Corporate Member of Freie Universität Berlin, Humboldt–Universität zu Berlin, and Berlin Institute of Health, BIH QUEST–Center for Transforming Biomedical Research, Berlin, Germany, **4** Charité–Universitätsmedizin Berlin, Corporate Member of Freie Universität Berlin, Humboldt–Universität zu Berlin, and Berlin Institute of Health, Institute of Biometry and Clinical Epidemiology, Berlin, Germany, **5** Berlin Institute of Health (BIH), Berlin, Germany, **6** Kliniken Beelitz GmbH, Beelitz-Heilstätten, Germany, **7** Department of Neurology, University Medicine Greifswald, Greifswald, Germany, **8** German Center for Neurodegenerative Diseases (DZNE), Partner Site, Rostock/Greifswald, Germany

☯ These authors contributed equally to this work.

* torsten.rackoll@charite.de

**Data Availability Statement:** Data and analysis scripts are available at https://doi.org/10.5281/zenodo.3718578.

## Abstract

### Introduction

Many clinical studies reporting accelerometry data use sum score measures such as percentage of time spent in moderate to vigorous activity which do not provide insight into differences in activity patterns over 24 hours, and thus do not adequately depict circadian activity patterns. Here, we present an improved functional data analysis approach to model activity patterns and circadian rhythms from accelerometer data. As a use case, we demonstrated its application in patients with mild cognitive impairment (MCI) and age-matched healthy older volunteers (HOV).

### Methods

Data of two studies were pooled for this analysis. Following baseline cognitive assessment participants were provided with accelerometers for seven consecutive days. A function on scalar regression (FoSR) approach was used to analyze 24 hours accelerometer data.

### Results

Information on 48 HOV (mean age 65 SD 6 years) and 18 patients with MCI (mean age 70, SD 8 years) were available for this analysis. MCI patients displayed slightly lower activity in the morning hours (minimum relative activity at 6:05 am: -41.3%, 95% CI -64.7 to -2.5%, p =

**Funding:** This work was supported by the Deutsche Forschungsgemeinschaft (DFG, German Research Foundation) – Project number 327654276 – SFB 1315 to AF.

**Competing interests:** The authors have declared that no competing interests exist.

**Abbreviations:** BDI, Becks Depression Inventory; CI, Confidence interval; d-MEQ, Morningness-Eveningness-Questionnaire (german version); ESS, Epworth Sleepiness Scale; FDA, Functional data analysis; FoSR, Function-on-scalar-regression; MCI, Mild cognitive impairment; MMSE, Mini-Mental-State-Examination; MRI, Magnetic Resonance Imaging; HOV, Healthy older volunteers; PANAS, Positive and Negative Affect Schedule; PSQI, Pittsburgh Sleep Quality Index; SMD, Standardized mean difference; STAIG-X1, State-Trait Anxiety Inventory (german version); SVF, Stress-Verarbeitungs Fragebogen (translated: stress coping questionnaire); tDCS, transcranial direct current stimulation; WHOQoL, World Health Organization Quality of Life.

0.031) and in the evening (minimum relative activity at 21:40 am: -48.4%, 95% CI -68.5 to 15.4%, p = 0.001) as compared to HOV after adjusting for age and sex.

## Discussion

Using a novel approach of FoSR, we found timeframes with lower activity levels in MCI patients compared to HOV which were not evident if sum scores of amount of activity were used, possibly indicating that changes in circadian rhythmicity in neurodegenerative disease are detectable using easy-to-administer accelerometry.

## Clinical trials

Effects of Brain Stimulation During Nocturnal Sleep on Memory Consolidation in Patients With Mild Cognitive Impairments, ClinicalTrial.gov identifier: NCT01782391. Effects of Brain Stimulation During a Daytime Nap on Memory Consolidation in Patients With Mild Cognitive Impairment, ClinicalTrial.gov identifier: NCT01782365.

## Introduction

Activity assessment in natural environment is a widely used approach to determine habitual differences between cohorts [1] and has been used in different disease populations [2, 3] or as a predictor for overall mortality [4]. Devices to detect activity of body movements include small inertial measurement units (IMUs). These accelerometers have the advantage that they capture bodily movement with a high precision in time within natural environments. Still, many studies using accelerometers in clinical populations report aggregated data on number of steps per day or on amount of time spent in high, low or no activity as well as number of bouts without activity [5]. The problem of aggregated data is that it reduces acquired information, thereby losing the high dimensionality of continuous 24 hours recordings and minimizing the information content of accelerometry. A more holistic view on activity distributions in daily rhythms has been postulated as an emerging topic lately [6]. Thus, more detailed assessments of biological rhythms are needed [7].

Time series analyses has been used to capture e.g. circadian rhythms in humans but its application on accelerometer data is still the exception in clinical trials. Common measures of circadian organization of activity patterns report the amplitude, mesor and acrophase (time of the peak within the fitted 24 h rhythm) of activity which highlights the extremes of daily activity but have a poor resolution in time [8]. Differences in circadian activity responses might be subtle so more precise tools are needed to analyze biological rhythms data. Researchers from Johns-Hopkins University [9, 10] and Columbia University [11] introduced the concept of functional data analysis (FDA) to accelerometry data. In a first methodological study Goldsmith et al. [11] demonstrated that while standard procedures showed inconclusive results, FDA was able to demonstrate differences in distribution of activity pattern over time. This approach thus might be able to detect subtle differences in circadian organization of activity patterns in a variety of disease states, including incipient neurodegenerative disease.

The pathology of Alzheimer's disease (AD) serves here as a promising clinical example as AD starts years or even decades before first clinical symptoms become apparent [12]. Thus, early diagnosis is mandatory to test approaches aimed to slow or even halt the progression of cognitive decline, including physical activity, cognitive training, dietary supplementation or

complex lifestyle interventions [13–15]. Several markers, including disrupted sleep behavior have been described before as a potential screening target [16]. Likewise a disruption of circadian activity rhythm has been proposed [17]. In early stages of neuronal degeneration reduction of the suprachiasmatic nucleus (SCN) has been observed [18]. The SCN has often been described as the central clock of the circadian network which controls melatonin and cortisol secretion and consequently influences alertness and sleep behavior [19, 20]. Likewise a disruption of circadian rhythmicity e.g. in the melatonin secretion cycle has been observed in patients with symptoms of neurodegenerative disorders [17, 20]. Daily rest-activity patterns are hypothesized to be controlled by circadian functions and its disruption has been observed in AD patients and in mouse models of AD [21–24]. Therefore, assessment of activity patterns might be well-suited to indicate AD pathology in observational studies. Moreover, such assessments with small IMUs would constitute non-invasive and low-cost approaches that may allow for screening of large cohorts. In addition, these tools could be employed for repeated follow-up in long-term preventive or therapeutic interventions like lifestyle changes or dietary supplementation.

Here, as a clinical example we applied a functional data analysis algorithm on healthy older volunteers (HOV) and patients with mild cognitive impairment (MCI) in a previously acquired data set to describe the circadian rhythm in both groups, and to test if their activity patterns differed throughout a 24 hour cycle. We hypothesize that our algorithm would be able to detect differences in distribution of activity with regard to timing and magnitude between HOV and MCI. Furthermore, we provide with detailed methodology in the S1 Appendix, and our analysis scripts are available online (see https://doi.org/10.5281/zenodo.3718578) to encourage future applications of this approach.

## Methods

Data reported here were taken from baseline measurements of two intervention studies that assessed the effects of oscillatory tDCS during sleep (daytime nap vs night sleep) on cognitive performance (Effects of Brain Stimulation During a Daytime Nap on Memory Consolidation (study 1) & Effects of Brain Stimulation During Nocturnal Sleep on Memory Consolidation (study 2) in young and older healthy subjects and subjects with mild cognitive impairment (MCI)). These studies are described in detail elsewhere [25, 26] and included comprehensive assessment of subjective (using sleep diary and sleep questionnaires) and objective sleep-wake behavior among others, as well as neuropsychological testing and structural imaging of the brain using magnetic resonance imaging (MRI). Objective sleep assessment included seven days of accelerometry. In the current analysis we used all available baseline data of patients with MCI and age-matched HOV.

### Study approval

The studies were approved by the institutional review board of the Charité Universitätsmedizin Berlin, Germany (EA1_295_12 & EA1_028_12), and were conducted in accordance with the declaration of Helsinki (Version 2008). All participants gave written informed consent prior to participation, and received a small reimbursement for their time.

### Participants

HOV (50–90 years) were recruited via advertisements in the local database of the Charité Universitätsmedizin Berlin, Germany. MCI patients (50–90 years) were referred to the study from the memory clinic of the Charité University Hospital. The original studies included also young healthy participants; for the current analyses we focused on HOV and MCI, given our aim to

compare circadian organization of activity in older adults. Participants underwent a structured telephone interview to exclude the presence of manifest sleep disturbances, contraindications for MRI, and non-native German speakers. Out of 242 potential participants that had entered the study, 66 HOV (mean age 66, SD 6) and 31 MCI (mean age 70, SD 8) were then invited to the laboratory to determine study eligibility which included a clinical interview, neurological examination, structural MRI, and standardized cognitive testing. Inclusion and exclusion criteria for both groups are detailed in the supplementary. In short HOV participants had to show no signs of dementia, and no present episode of depression (monitored with the Beck's Depression Inventory (BDI [27])) while MCI patients had to fulfill core clinical criteria for the diagnosis of MCI outlined by Petersen and others [28]. Further, clinical assessment and structural MRI revealed no systemic or brain diseases accounting for declined cognition. Patients diagnosed with amnestic or amnestic plus MCI were included. All these criteria had to be fulfilled by the patients for inclusion in the original studies and only eligible patients were targeted for accelerometry assessment.

Both groups were assessed using a neuropsychological test battery addressing various cognitive functions to ensure that HOV performed within age and gender matched normal range. A detailed description of all domains and tests can be found in the S2 Table in S1 File.

## Accelerometry

The ActiGraph GT3X+ (ActiGraph, Pensacola, FL, USA) was used in this study. It is able to assess acceleration in the vertical, antero-posterior and medio-lateral axes. It has shown high inter-instrument reliability (Intraclass correlation 0.97 [29]) and intra-instrument reliability within frequencies that are common in human activities, and is described as a reasonable tool for longitudinally measuring sleep [30]. During the week following baseline cognitive assessment each participant wore a GT3X+ on the hip S2 Fig in S1 File for continuous seven-day recordings to fully capture daily physical activity. Subjects were asked to wear the device at all times and just to remove it for any activity involving water (showering, swimming, etc). The devices were pre-programmed with default settings (30 Hz in three axis, with a fixed start and stop time). Data were downloaded in 60 sec epochs. Accelerometry data download and sum score descriptives were performed using ActiLife Software 6.8.2 (ActiGraph, Pensacola, FL, USA). Following cut-points for sum scores of time spent in levels of activity were used: We picked 0–99 activity counts per minute as sedentary activity, 100–2019 as light activity, 2020–5998 as moderate and everything above 5999 as vigorous activity [31] as these cut-points derived from a large NHANES cohort of 6329 participants and thus were assumed to correctly reflect levels of activity [32]. We performed a multiple linear regression with average activity count as a dependent variable and group as an explanatory variable adjusted for age and sex.

Sleep scoring is based on the amount of activity counts per epoch and on the directionality of the sensors. Sleep onset and awakening time were rated visually by observation of a sharp decrease or increase of signal by two independent and experienced raters and controlled with sleep diaries administered to the patients before accelerometry assessment [33]. For automated sleep scoring we used the Cole-Kripke algorithm within the ActiLife software. The software then automatically calculates total time in bed, total sleep time, sleep latency, number of arousals, average duration of awakening and wake after sleep onset.

## Statistical analyses

We applied a Function-on-Scalar-Regression (FoSR) approach that is in line with recent publications by Goldsmith et al. [11] and the group of Biostatistics from Johns Hopkins University [9, 10, 34]. Function on scalar in this study means that activity counts as main outcome are

functions of time (response), while covariates, here MCI patients/HOV etc. are scalars. However, for our analysis we made two important modifications:

- The set of basis functions used in the regression model are discrete wavelet functions instead of cubic splines (For comparison see S3 Fig in S1 File) [7].

- In addition to the fixed effects, the model equation contains a random intercept to account for clustering of data within individuals.

Pre-processing steps for FoSR as well as all statistical analyses were performed using statistical software R (Version 3.2.1) and the R package 'wavelets' (version 0.3–0) [35].

We summarized the activity data using 5 minutes epochs resulting in 288 count data sets per participant per day (24h = 1440 minutes = 288*5 minutes) [10]. The rationale for the 5 minutes epoch was that we aimed to identify group differences of activity levels over the course of a day, focusing on identifying periods with high/middle/low physical activity, respectively. Furthermore, we log-transformed activity counts as distribution of count data were skewed. For high resolution accelerometer data robustness of data is a problem as changes between zero activity and high activity peaks might be present within small time periods. Furthermore, rapid changes in activity might not reflect circadian rhythm activity patterns. Therefore, a smoothing algorithm is needed to distinguish between signal (circadian activity pattern) and noise (changes between zero activity and high activity peaks within small time periods). Cubic splines or wavelet transformation are both suggested for smoothing of this type of data [7, 36]. Discrete wavelet functions and cubic splines have both a localized support, meaning that each wavelet function / each piece of the cubic spline model is able to fit the activity data well in a particular time range and reduces noise in this time range. In contrast to cubic splines, wavelets are periodic basis functions (with a period of 24 hours) and are thus better suited for modelling circadian patterns. Here discrete wavelet transformation was used for smoothing activity counts.

**Discrete wavelet analysis.**   The discrete wavelet transformation is a well-established method in the analysis of time series (c.f. [37] and [38] pp. 174–179). Different series of wavelets are in use for the analysis of time-series (Coiflets, Least Asymetric, Best Localized, Daubechies wavelets, all implemented in the R package 'wavelets'). The full discrete wavelet transformation is a regular linear (matrix) transformation from a finite dimensional real vector space into a space of the same dimension. Here, we used incomplete wavelet transformation with a non-trivial kernel that maps the time series from a high dimensional space (dim = 288) into a real space with far less dimensions (dim = 18, 18 basis functions). The purpose of this approach is twofold:

- Smoothing the activity data (removing noise).

- Defining a set of basis functions which can be used in the function-on-scalar regression analysis for testing group/subgroup specific activity patterns.

The incomplete discrete wavelet transformation behaves like a low-pass filter, meaning that rapid changes between non-activity and high activity will be smoothed out. For our analysis, we chose Daubechies wavelets of length 10 (d10, which corresponds to a time range of 50 minutes for the first level wavelet functions), meaning that an activity pattern of 50 minutes is modelled by one function.

The degree of smoothing depends on the number of basis functions used, which results from the length of the functions. Shorter wavelet filters tend to give poor results in smoothing the data (not enough smoothing, too much noise) whereas wavelets with long filter length are not well localized (too much smoothing, no sufficient model fit with regard to pattern relevant

changes in activity). The use of 18 basis functions seemed to be a good compromise between good smoothing results and preserving a sufficiently high resolution for detecting activity patterns within circadian rhythm. The incomplete wavelet transform in concise matrix notation is

$$x_{sm} = Y_{lp}Y_{lp}^t x$$

$x$ denotes the vector containing the raw data whereas $x_{sm}$ is the vector of smoothed data (the subscript $sm$ stands for "smoothed"). The columns of the matrix $Y_{lp}$ are the 18 basis function of the incomplete wavelet transform. Since each basis function is a vector of length 288, the matrix $Y_{lp}$ has 288 rows and 18 columns. Finally, the subscript $lp$ refers to the low pass property of the incomplete wavelet transform.

**Function-on-scalar-regression model.** The presentation of the technical details of Function-on-Scalar-Regression closely follows the presentation in the appendix of [37]. The model equation of FoSR is largely analogous to the model equation of a linear mixed effects model. We included a random intercept in the model equation since some participants contributed several correlated 24 hours activity records to the study.

Let $x_{ij}$ be the vector of a log(1+count) transformed 24 hour activity record for day $j = 1,\ldots,j_i$ of participant $i = 1,\ldots,N$ aggregated in 5 minutes time blocks. Since each volunteer wore the actigraph device between one and eight days, $j_i$ may range from 1 to 8. Hence, the model equation is

$$x_{ij}(t) = b_i + \beta_0(t) + \sum_{k=1}^{p} \beta_k(t)\xi_{ik} + \varepsilon_{ij}(t); \ i = 1,\ldots,N; \ j = 1,\ldots,j_i; \ t = 0,\ldots,287. \quad [1]$$

In Eq [1] $\xi_{ik}$ denotes the value of the $k^{th}$ covariate of the $i^{th}$ participant and $\varepsilon_{ij}(t)$ are the normally distributed residual terms with expectation zero and common variance $\sigma^2$. The random intercept terms $b_i$ for the participants are normally distributed with expectation zero and variance $\tau^2$. We expanded each (row) coefficient vectors $\beta_k(t)$ using the wavelet basis functions that are the columns of $Y_{lp}$:

$$\beta_k = \gamma_k Y_{lp}^t \ k = 0,\ldots,p \quad [2]$$

Omitting the index $t = 0,\ldots,287$, the vector $\beta_k$ has length 288 and $\gamma_k$ is a row vector of length 18 (number of wavelet basis functions). Details of the estimation of model parameters are presented in the appendix.

The FoSR model provides for each covariate (e.g. a group indicator variable) absolute and relative differences of activity counts with 95% confidence intervals and p-value for each five minutes epoch. We report periods where the covariate (e.g. an indicator variable defining group differences) is significant in all five minutes époques and report only the largest relative difference in percent with two-sided 95% CI and p-value. All analyses were controlled for age and sex in order not to overfit the model following the suggestions of Xiao et al. that the number of subjects included in the model should steer the decision on model size and complexity [11].

**Addressing the problem of multiple testing.** All p-values were Bonferroni adjusted by a factor of 18. The Bonferroni factor was chosen equal to the number of basis functions used in the FoSR model. Accordingly, the confidence level of the confidence bands was risen from 1-α to 1-α/18.

## Results

97 HOV and MCI patients were available for inclusion (patient flow is displayed in S1 Fig in S1 File). Of these 97 participants, 2 HOV violated the inclusion criteria of the initial studies

and were consequently excluded from activity assessment. Twenty-four participants (14 HOV and 10 MCI patients) had to be excluded due to missing actigraph data for technical or logistic reasons. An additional five participants (2 HOV and 3 MCI patients) had to be excluded due to incoherent zero values distributed over a 24 hours cycle at all study days. Here, zero values distributed within valid measurements distorted the smoothing algorithm. Thus 48 HOV (mean age 65, SD 6) and 18 MCIs (mean age 70, SD 8) remained for final analysis with 232 valid accelerometer days in HOV and 78 days in MCI respectively. We compared basic characteristics of included and excluded subjects but could not identify major differences.

## Baseline characteristics

Information on baseline characteristics including sociodemographic characteristics and cognition can be found in Table 1. Overall, MCI patients differed from HOV with regard to age, (mean 70 vs. 65 years, SMD = 0.58), depressive symptoms (median 7 vs. 3 in BDI, SMD = 0.88) and in cognitive functioning (mean MMSE 28 vs. 29, SMD = 0.90).

Participants spent most of their day in a sedentary lifestyle or in light activity with only small amounts of time in higher activity levels (Table 1), lacking substantial differences between groups (SMD for all measures <0.5). Based on the simple linear regression model, average activity count was similar in both groups (-7 counts for MCI compared to HOV, 95% CI -20 to 7, p-value = 0.32). Aggregated analyses of sleep assessed via accelerometry exhibited

**Table 1. Baseline characteristics and aggregated descriptive accelerometer data for HOV and MCI.**

| | HOV (n = 48) | MCIs (n = 18) | SMD standardized mean difference |
|---|---|---|---|
| Female sex, no (%) | 25 (52) | 10 (56) | 0.07 |
| Age in years, mean (SD) | 65 (6) | 70 (8) | 0.58 |
| Education in years, mean (SD) | 16 (3) | 16 (5) | 0.12 |
| MMSE, mean (SD) § | 29 (1) | 28 (1) | 0.90 |
| Becks Depression Inventory, median (IQR) | | 3 (1–4) | 7 (5–11) | 0.88 |
| **Sleep scoring** | **HOV** (n = 48) | **MCI** (n = 18) | **SMD standardized mean difference** |
| Time in bed, mean (SD) | 23:20 (77) | 23:01 (71) | 0.11 |
| Time out of bed, mean (SD) | 07:41 (81) | 07:22 (54) | 0.26 |
| Total time in bed (min), mean (SD) | 497 (62) | 501 (79) | 0.06 |
| Total sleep time (min), mean (SD) | 484 (62) | 487 (75) | 0.04 |
| Wake After Sleep Onset (%), mean (SD) | 2.5 (1.5) | 2.6 (1.7) | 0.06 |
| Arousals (no.), mean (SD) | 4.3 (2.4) | 3.7 (2.4) | 0.23 |
| Average Awakening (min), mean (SD) | 3.2 (1.0) | 3.9 (1.4) | 0.54 |
| Sleep latency (min), mean (SD) | 1.8 (1.1) | 1.6 (1.0) | 0.19 |
| **Physical activity** | **HOV** (n = 48) | **MCI** (n = 18) | **SMD standardized mean difference** |
| Energy Expenditure (kcal/day), mean (SD) | 412 (176) | 333 (265) | 0.35 |
| Average activity count per day, mean (SD) | 426 (139) | 396 (169) | 0.19 |
| Time in sedentary activity (%), mean (SD) | 74 (8) | 72 (7) | 0.32 |
| Time in light activity (%), mean (SD) | 23 (6) | 25 (5) | 0.46 |
| Time in moderate activity (%), mean (SD) | 2 (1) | 3 (4) | 0.13 |
| Time in vigorous activity (%), mean (SD) | 0 (0) | 0 (0) | 0.49 |
| Moderate-to-vigorous physical activity (%), mean (SD) | 3 (2) | 3 (4) | 0.08 |

§ MMSE denotes mini mental state examination and is a score ranging from 0 to 30 screening for overall cognitive functioning in which higher values describe better cognitive functioning.

| 4 missing at random (3 in HOV, 1 in MCI)

Two-tailed t-test or Wilcox signed rank test were used for metric data and Chi squared statistics for categorical data.

comparable characteristics between the two groups (most SMD 0.2 or lower) only for average awakening MCIs had somewhat higher values (SMD = 0.54).

## Function-on-scalar-regression

We compared the distribution of activity in MCI patients with HOV over the averaged 24 hours day-night cycle using FoSR. Variability of activity levels was higher in MCI patients compared to HOV. Fig 1A displays the smoothed average absolute activity over the course of a 24 hour cycle. MCI showed higher activity levels throughout the waking hours with a slight decrease below levels of HOV in the evening hours. For direct comparison between groups we analyzed relative activity between MCI and HOV adjusted for age and sex (Fig 1B). We observed lower relative activity in MCI patients compared to HOV in the morning hours between 5:40 and 6:20 am with the lowest value at 6:05 am (-41.3%, 95% CI -64.7 to -2.5%, p = 0.031). In the evening hours between 9:05 and 10:25 pm MCI patients showed a relative lower activity with its lowest value at 9:40 pm compared to HOV (-48.4%, 95% CI -68.5 to 15.4%, p = 0.001). Over the course of the sleeping hours (mean time to bed and time out of bed hours in each group are displayed in Table 1) activity patterns of MCI patients and HOV were comparable. In sum MCI patients were less active in the morning and in the evening hours compared to HOV. Activity levels by time for two sample participants with fitted curves from the FoSR is shown in Fig 2.

## Discussion

In this study our goal was to apply the previously published Function-on-Scalar-Regression (FoSR) algorithm for the analysis of accelerometry data to a clinical population to provide an

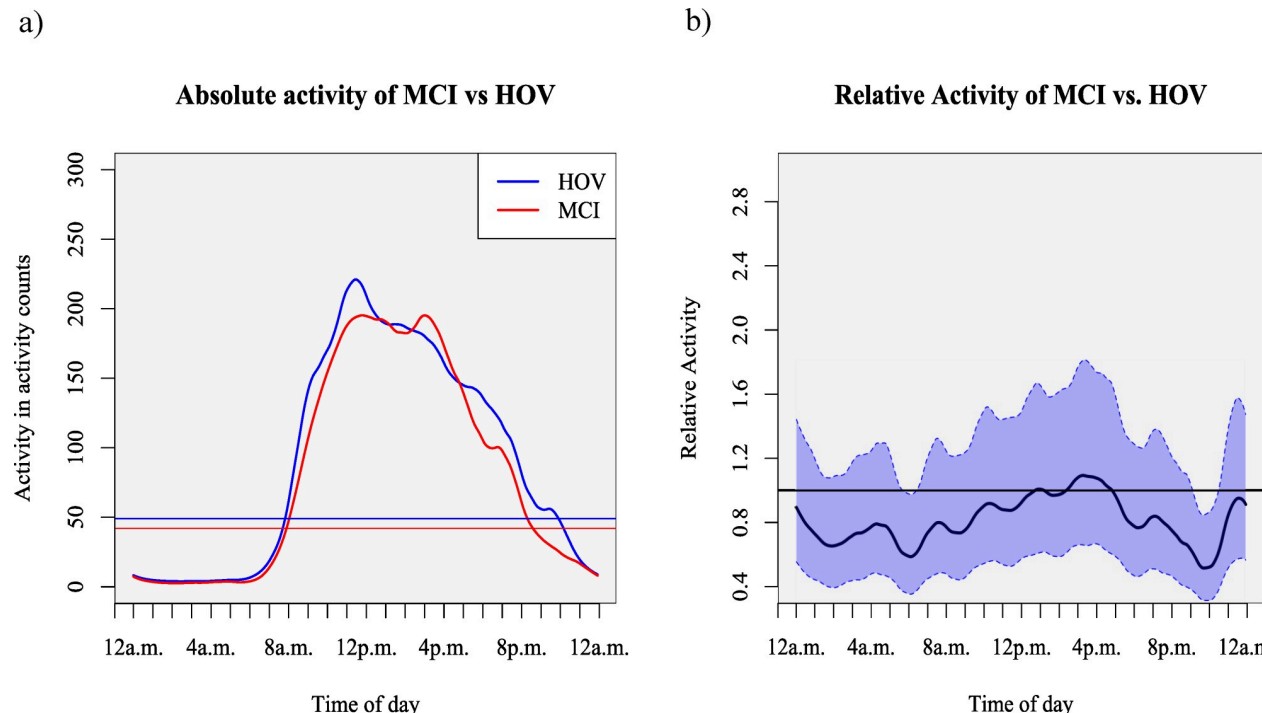

a)                                                                 b)

**Fig 1. Comparison of absolute and relative activity distribution between MCI and HOV.** Distribution of a) mean absolute activity between MCI and HOV over time and b) mean relative activity between MCI and HOV over time with 95% Confidence Interval (CI) adjusted for age. Horizontal lines in Fig 1A) displays the average daily activity of log-transformed data from fitted curves of respective groups. The black line in Fig 1B) displays a relative activity. A value of one indicates no difference of activity between the two groups.

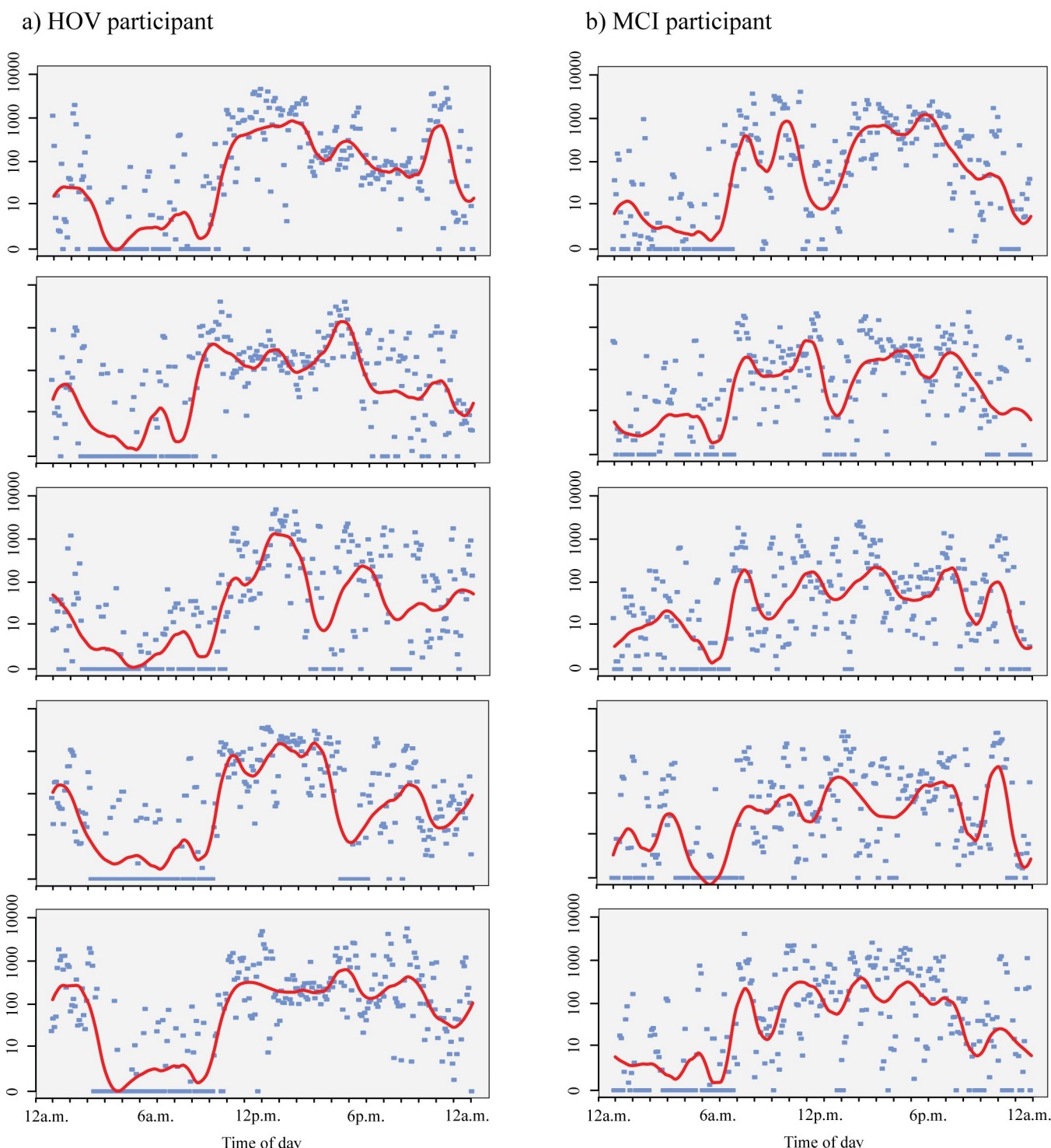

**Fig 2. Activity level by time for two sample participants with fitted curve from FoSR.** Distribution of absolute activity over a five day period for A) one HOV sample participant and B) one MCI sample participant. Red lines denote the wavelet smoothing.

example for the algorithm's potential in observational or intervention studies. We analyzed data from two pooled, previously published trials with the commonly applied approach in accelerometry, i.e., accelerometer derived sum scores [5, 39]. Here, we did not find substantial differences in proportions of time spent in specific activity levels or in total volume of activity. Next, we administered FoSR to investigate the circadian organization of activity patterns in patients with MCI and HOV using time series analysis and found lower relative activity in the morning and in the evening hours in MCI patients compared to HOV. While the net difference in amount of activity is likely to be negligible, our approach allowed us to demonstrate different time-based distributions of activity between the two populations.

Addressing circadian organization of activity, several authors have used five or ten hours spent in lowest or highest activity to assess phase shifts in activity in patients with MCI or dementia [22, 40, 41]. Overall, findings from these studies have not been unequivocal, which indicates that especially in the early stages of cognitive impairment differences might be subtle. Musiek et al. [22] described in their study differential activity patterns distributed throughout the day with lower amplitude between lowest and highest amount of activity and an increased variability over the course of 24 hours in preclinical AD patients. The authors compared different methods to analyze accelerometer data but did not apply time sensitive functional analysis. In contrast, time sensitive functional analysis was central in our study to identify time zones during which MCI patients were more or less active compared to HOV.

Further, a disruption of circadian activity synchrony has been demonstrated in mice models of AD [24, 42] and been described in human AD cohorts [43, 44]. Our study cannot determine if the differences found between HOV and patients with MCI would lead to earlier detection of MCI in screening of individuals in the community, but this issue can now be addressed in future large-scale studies.

## Limitations

Several limitations should be considered when interpreting our findings. First, groups of HOV and MCI patients differed with regard to age, which might introduce a bias due to altered sleep behavior in the process of aging. However, we believe that these differences did not substantially impact our main results, given that the analysis was controlled for age. Second, the sample of our MCI group was small compared to the HOV group. However, on average we used data of four days per participant which provides a stable estimation of the participant's daily activity pattern. Also, we used our data as a test case to demonstrate an analysis approach with high potential in a wide range of future applications. The validity of this approach has to be confirmed in future larger trials.

## Future research

First, replication in large longitudinal studies are needed to corroborate our results, and to evaluate the potential of actigraphy with FoSR analysis as a screening tool for incipient neurodegenerative disease. Also assessment in different pathological and healthy conditions are warranted to extract specific alterations in activity patterns for a certain pathology like AD. Second, results derived from accelerometers with a FoSR approach should be compared to wrist-worn accelerometers, to facilitate studies of larger cohorts with commercially available activity monitors. Third, to inform inferences on the underlying mechanisms, biological markers such as melatonin and cortisol secretion or body temperature should be determined to more accurately define circadian rhythms. Forth, model choices in FoSR such as number of basis functions or correction methods for multiple testing should be validated in different patient populations to enhance the informative value of the regression method. Finally, it

should be assessed if parameters described here are sensitive to changes over time in response to therapies.

## Conclusion

Using a novel approach of FoSR, we were able to demonstrate improved precision in analysis of time based high dimensional activity data. We found that MCI patients exhibited decreased relative activity in the morning and in the evening hours compared to HOV. The approach of the FoSR to analyze activity patterns may constitute an important screening tool for incipient neurodegeneration in the population at large, particularly if accelerometers of commercial smart watches are found to provide with valid measures for FoSR.

## Supporting information

**S1 File.**
(DOCX)

## Acknowledgments

We thank Lena Reich for their help with data acquisition. Special thanks to Patrizia Müller and Madlee Einsiedler for data preparation and actigraphy sleep ratings. Additionally, we want to thank Luo Xiao, Jeff Goldsmith and Ciprian Crainiceanu for provision of their R-scripts and helpful comments on the implementation of functional data analysis in accelerometry data. Furthermore, we would like to thank all subjects for their participation.

## Author Contributions

**Conceptualization:** Torsten Rackoll, Ulrike Grittner.

**Data curation:** Torsten Rackoll, Sven Passmann.

**Formal analysis:** Torsten Rackoll, Konrad Neumann, Ulrike Grittner.

**Funding acquisition:** Nadine Külzow, Agnes Flöel.

**Investigation:** Sven Passmann, Julia Ladenbauer.

**Methodology:** Torsten Rackoll, Konrad Neumann, Ulrike Grittner.

**Project administration:** Nadine Külzow, Agnes Flöel.

**Software:** Torsten Rackoll, Konrad Neumann.

**Supervision:** Ulrike Grittner.

**Visualization:** Konrad Neumann.

**Writing – original draft:** Torsten Rackoll, Konrad Neumann.

**Writing – review & editing:** Sven Passmann, Ulrike Grittner, Nadine Külzow, Julia Ladenbauer, Agnes Flöel.

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
