## [Decision Letter · Decision Letter 0]

9 Jun 2020

PONE-D-20-08099

Applying time series analyses on continuous accelerometry data – a clinical example in older adults with and without cognitive impairment

PLOS ONE

Dear Dr. Rackoll,

Thank you for submitting your manuscript to PLOS ONE. After careful consideration, we feel that it has merit but does not fully meet PLOS ONE’s publication criteria as it currently stands. Therefore, we invite you to submit a revised version of the manuscript that addresses the points raised during the review process.

Pls. very carefully address the major concerns by the reviewers, in particular with reagrd to cohort sizes, statisitcis, methodology etc.

We look forward to receiving your revised manuscript.

Kind regards,

Henrik Oster, Ph.D.

Academic Editor

PLOS ONE

Journal Requirements:

'This work was supported by grants from the Deutsche Forschungsgemeinschaft (Fl 379-10/1;

Fl 379-11/1, and DFG-Exc 257).'

'The funders had no role in study design, data collection and analysis, decision to

publish, or preparation of the manuscript.'

Additional Editor Comments (if provided):

Dear Dr. Rackoll,

as you can see from the reviews several fundamental concerns were raised regarding the validity of your study. Please very carefully address all the major concerns (such as sample size, statistics, methodology) in your revision.

Reviewers' comments:

Reviewer's Responses to Questions

**Comments to the Author**

1. Is the manuscript technically sound, and do the data support the conclusions?

Reviewer #1: Yes

Reviewer #2: No

Reviewer #3: No

2. Has the statistical analysis been performed appropriately and rigorously? 

Reviewer #1: Yes

Reviewer #2: I Don't Know

Reviewer #3: No

3. Have the authors made all data underlying the findings in their manuscript fully available?

Reviewer #1: Yes

Reviewer #2: No

Reviewer #3: No

4. Is the manuscript presented in an intelligible fashion and written in standard English?

Reviewer #1: Yes

Reviewer #2: Yes

Reviewer #3: Yes

5. Review Comments to the Author

Reviewer #1: General: The authors are applying a relatively novel analysis, function-on-scalar regression, to differentiate changes in diurnal activity between older adults with and without MCI. The authors note that this method is an improvement from aggregated approaches of determining links between activity and covariates such as demographic and health measures. Overall, this is a sound study and is utilizing an appropriate model to extend our knowledge of activity based measures in the presence or absence of disease in older adults.

Major Points

The type of basis function used may improve or hurt prediction accuracy. You should compare model accuracy of different basis functions through either cross-validation for each type of basis function or inspection of the residual curves produced by the model and then determine which is the better fit.

On line 206 you state that data should be aggregated into at least 5 minute epochs which is what you chose moving forward. What is the rationale for not choosing a smaller epoch of less than 5 minutes?

On line 243 you state that 18 basis functions seemed to be a good compromise between smoothing out noise versus preserving activity patterns. Since there is little previous research to support this methodology the determination of number of basis functions should be driven by model comparison (i.e. comparison of residual curves produced by each model to detect bias in prediction as a result of number of basis functions). It is difficult to know if 18 basis functions is best without any tests of cross validation or other model comparison metrics.

What are the results in table 1 reported with an unadjusted p-value? Isn’t each row of this table an independent test and thus inflating your chance of a type 1 error?

In reviewing the cited Goldsmith paper, https://www.ncbi.nlm.nih.gov/pmc/articles/PMC4987214/, shouldn’t the Function-of-Scalar-Regression (FoSR) give you a global coefficient and p-value for each covariate used in your model? The stated purpose of this paper should be to compare results of multiple linear regression using summer activity levels compare to results of the FoSR in terms of global beta and p-values, just like that referenced Goldsmith paper. See table 1 from the linked study above. This would fulfill the primary purpose of this paper to demonstrate the sensitivity of the FoSR analysis compared to more traditional regression approaches.

Also, does your FOSR model only include covariates of group (MCI vs HOV) and age? In line 279 you state that you only control for age to prevent overfitting but how do you know that you have overfit the model in this context without doing some level of cross validation which is absent here?

And again in reference to the previous Goldsmith paper when you are plotting relative activity in figures 1 or 2 is this similar to their plotting of the coefficient function (i.e. the change in the prediction of activity level as a function of the covariate across time)? See figure 1 and 2 in Goldsmith 2016. I continue to bring up the Goldsmith paper because it would be in the best interest of this paper to demonstrate the strength of this method based on how it has been used previously to help identify differences and similarities in your approach and interpretation compared to how it was used previously.

In figure 1 you should demonstrate how the predicted values of the FoSR model differ from a simple averaging of the data in both the absolute and relative activity counts between groups. This would provide more face validity on the conceptual basis of how FoSR captures activity patterns in this sample of data that averaging and summing methods do not.

In the chronotype analysis are you still using FoSR to determine differences in activity level as a function of strata and group? It does not explicitly say if this is the case. I believe you say it on line 404 but it should be stated earlier.

You may also wish to include the experiment that each individual comes from as a random variable/effect which may also help control for any variance in this study since this sample is a pooled group from two previous projects.

Minor Points:

Line 258 do you mean that you inputted multiple 24-hour activity records from one individual into your model?

Line 276 change epoque to epoch

Line 309 should OLD be HOV?

Line 335 change hours to hour

In figure caption for Figure 1 include what the black line in B represents.

Reviewer #2: The data and question could be used to write a useful paper but this one tries to do too much with too little and isn't careful about how it is done. The motivation is not appropriate for the question, the paper contains irrelevant material, the analytic sample is tiny and extremely selected, so I place no confidence in the results.

Reviewer #3: Rackoll and colleagues describe the results of a “… time series analyses on continuous accelerometry data – a clinical example in older adults with and without cognitive impairment” and find that patients with mild cognitive impairment show an elevation in activity levels, specifically in the morning and the afternoon.

The first sentence of the MS is: “Current analysis approaches of accelerometry data use sum score measures which do not provide insight in activity patterns over 24 hours, and thus do not adequately depict circadian activity patterns”.

What are the authors talking about? What is meant by “Current analysis approaches”? Do the authors refer to commercial applications that come with the accelerometers, or the read-outs of smart watches?

Whatever is meant by this introductory sentence, it is far from reality and shows that the authors are not familiar with the field of circadian research. Time-series analysis of circadian data has been uses decades before even accelerometers were applied in human studies, for example by Sokolove in the 1970s. But even today and specifically concerning actimetry in humans, the search – “time series analysis" circadian human actimetry – yields more than 350 papers, 16 alone in 2020. The top hit in relevance is a paper called “Multiscale adaptive analysis of circadian rhythms and intradaily variability: application to actigraphy time series in acute insomnia subjects”, which even has been published in PLOS one. I am willing to bet that all of these papers go beyond “sum score measures”. The authors should look at this paper as an example of what can be done when analysing “continuous accelerometry data”.

It seems that the authors use FoSR similar to how some people use SPSS, i.e., without actually looking at the data. The figures the authors come up with are simple 24-hour averages as have been used in analyses of human actimetry for decades. It is simply not enough to state that there are activity level-differences between the groups. Actimetry can do so much more, can dig so much deeper than the analysis presented here. The first step of every detailed analysis of circadian long-term time series are making double plots, so that one can see how the activity is generally spread over every day’s 24 hours. The PLOS one paper mentioned above, goes into “intradaily variability”, which could well be at the basis of the activity level differences. We have no idea how dispersed or consolidated the daily activity profiles are in the participants of the two groups, we have no idea when the participants apparently sleep (which could be assessed by looking at the data).

Figure 1a should be also double-plotted, since this allows a proper representation of sleep. Even a relative short recording of a week allows to determine a mid-trough time of something like time of “L5” (often used in time series analysis of human actigraphy and mentioned in the present MS). These individual phases would have been much more appropriate to be used in a normalisation process compared to the MEQ questionnaire. It is also not clear whether individual daily profiles were normalised to individual mean levels (deviation from daily means or entered moving means). If one wants to make statutes about specific times of day where activity levels differ between groups, such a step would be essential.

If the only result that came out of this “novel” FoSR analysis is the vague activity level difference that can be sen in Fig 1a, then the field does not need this analysis. This is a shame because we need much more good analysis of actimetry data from every kind of patients, especially in psychiatry and for higher age-groups.

I suggest the authors team up with people, who have long-term experience in circadian analysis of activity recordings and milk their data again with all the insights available for this type of analysis.

6. PLOS authors have the option to publish the peer review history of their article (what does this mean?). If published, this will include your full peer review and any attached files.

Reviewer #1: Yes: Andrew Hooyman

Reviewer #2: No

Reviewer #3: No

---

## [Author Response · Author response to Decision Letter 0]

23 Nov 2020

Editor’s comments:

C#1: 

I was worried in particular about the loss of MCI patients because of “incoherent zero values.” Perhaps these are a result of cognitive limitations, which is exactly what you are trying to study. The sample was reduced also by “insufficient wear time”, which could also be due to cognitive issues. These problems with selectivity of the sample, with no attempt to characterize it or correct it, make the results suspect.

R#1:Thank you very much for addressing this important point as to the selection criteria. 

In our manuscript we excluded complete patient data if wear time was deemed insufficient over the entire 7-day period, as suggested by Choi and colleagues and several other studies.(2) Here, our goal was to attain datasets comparable with the literature. We do acknowledge that selection of data should be kept at a minimum or reasons for selection discussed. 

First, we now reevaluated patient data that did not meet the threshold of 60% wear time over the entire seven-day assessment. Within the data we found “complete days of accelerometer wear time” in eight patients (2 HOV and 6 MCI). We decided to include those patients as well. In other words, we now included all patients with at least one full day of accelerometer data but of those only complete days were used. With those additional data, we now analyzed 66 participants’ data of which 48 are HOV (73% of initially included HOV) and 18 are MCI (58% of initially included MCI). Please see our revised flow chart:

Regarding the reasons for the selection, we would briefly summarize how patients were selected. 24 participants of the initial 97 participants were excluded because no acceleration data was available (14 HOV and 10 MCI). Missing data was due to logistic or technical reasons and were not caused by the participants themselves. We excluded only five patients (2 HOV or 3 MCI) due to incomplete individual days or incoherent zero values. Here, we cannot identify worse adherence within the MCI patient group. Also, the relative contribution of individual recording days is approximately even between the two groups (4.8 days per HOV vs. 4.3 days per MCI patient). Therefore, on the basis of our data we see no systematic bias towards the selection criteria. We do acknowledge that a systematic bias could be problematic as to the inference taken from the data. Consequently, we rephrased the limitation which know reads as follows:

“Second, the sample of our MCI group was small compared to the HOV group. However, on average we used data of four days per participant which provides a stable estimation of the participant’s daily activity pattern. Also, we used our data as a test case to demonstrate an analysis approach with high potential in a wide range of future applications. The validity of this approach has to be confirmed in future larger trials.”

We would like to take the opportunity to provide more details on some technical problems with accelerometer data which might help the reviewer to appreciate our decision to exclude individual recording days due to incoherent zero values: 

Zero values in between valid measurement periods are a frequent issue which relates to the sampling frequency and the epoch lengths as well as to possible measurement errors of the devices. A high amount of zero values within valid measurements did in our case distort the smoothing algorithm. One possible solution would be to impute the missing values as suggested already by Lee et al.(3) Still, it remains uncertain whether a zero count is a valid measurement or if it is the result of not wearing the accelerometer device. That makes the application of any imputation algorithm questionable since it could be a source of considerable bias towards higher activity. Using only complete daily activity records diminishes the sample size and is, of course, also a possible source of bias. We therefore excluded only individual recording days for which smoothing was not possible. Distribution of included recording days were as follows:

Table 1: number of recorded days by group

 Number of recorded days

 1 2 3 4 5 6 7 8

Group 

HOV 1 2 5 8 15 15 1 1

MCI 1 2 2 2 7 4 0 0

We revised the manuscript accordingly and the results section now reads as follows:

“97 HOV and MCI patients were available for inclusion (patient flow is displayed in Figure S1 of the appendix). Of these 97 participants, 2 HOV violated the inclusion criteria of the initial studies and were consequently excluded from activity assessment. Twenty-four participants (14 HOV and 10 MCI patients) had to be excluded due to missing actigraph data for technical or logistic reasons. An additional five participants (2 HOV and 3 MCI patients) had to be excluded due to incoherent zero values distributed over a 24 hours cycle at all study days. Here, zero values distributed within valid measurements distorted the smoothing algorithm. Thus 48 HOV (mean age 65, SD 6) and 18 MCIs (mean age 70, SD 8) remained for final analysis with 232 valid accelerometer days in HOV and 78 days in MCI respectively. We compared basic characteristics of included and excluded subjects but could not identify major differences.” 

Please also note respective changes in the baseline characteristics in table 1 in the manuscript.

C#2: The paper also is motivated by assessing an approach that “might be able to detect subtle changes” but the data are not change data. They are differences in the cross section.

R#2: Thank you very much for giving us the opportunity as to be clearer on the motivation of the manuscript. The sentence cited above is part of the paragraph in which we introduce the FoSR algorithm and address one of its possible applications in clinical research using interventional or longitudinal study designs. We are aware that data provided here are not change data but describe two populations in a cross-sectional design. We rephrased our motivation and the description of the FoSR method. 

The introduction now reads as follows:

“This approach thus might be able to detect subtle differences in circadian organization of activity patterns in a variety of disease states, including incipient neurodegenerative disease.” 

We also rephrased other sentences that might be misinterpreted as to the nature of the data. The following sections of the discussion have now been rephrased: 

“Overall, findings from these studies have not been unequivocal, which indicates that in the early stages of cognitive impairment, differences might be subtle.”

“First, replication in large longitudinal studies are needed to corroborate our results, and to evaluate the potential of actigraphy with FoSR analysis as a screening tool for incipient neurodegenerative disease.”

C#3: It also points to early detection as valuable to allow treatments, but no treatments for cognitive decline currently exist.

R#3: We thank the editor to clarify our introduction as to the value of the FoSR in the research of cognitive impairment and dementia. One of the motivations to use data from patients with mild cognitive impairment is to support early detection of people at risk of developing dementia to allow early interventions to delay the clinical manifestation of the disease state or to slow progression of memory loss. While the reviewer is correct that no intervention is currently unequivocally recommended, recommendations with regard to cardiovascular risk factors and lifestyle decisions do exist (4), and several interventions are being investigated to slow progression of cognitive functioning, including physical activity (5), cognitive training (6), or dietary supplementation (7) to name just a few. Of note, cognitive impairment can be due to depression (8), or sleep disorders (9) which would result in specific treatment options. 

To underline the importance of assessing people at risk we added examples of approaches currently under study to delay further cognitive decline. The introduction now reads as follows: 

“Thus, early diagnosis is mandatory to test approaches aimed to slow or even halt the progression of cognitive decline, including physical activity, cognitive training, dietary supplementation or complex lifestyle interventions. (13 – 15)”

C#4: The Introduction promises hypotheses about differences in activity patterns for older adults with normal cognition vs those with MCI but presents none. It does not discuss the mechanisms through which cognitive limitation might affect activity patterns. 

R#4: Thank you for raising this issue, and thus allowing us to elaborate here. Our hypothesis on differences in activity patterns between MCI patients and aged-matched healthy older volunteers derives from clinical observations of changes not only in processing speed and mental flexibility (10) but also in activity behavior in individuals developing Alzheimer’s disease (AD) (11,12). Moreover, the relation between cognitive impairment and changes in acrophases of activity has been observed in healthy older adults (13). Similarly, changes in locomotor activity has been observed in a mouse model of AD, as compared to control animals. The changes described in the mouse model could be linked to altered clock gene expressions mostly in the hippocampus and the medulla-pons brain region (14). Interestingly, the authors did observe these changes in young animals and argue that these changes would precede amyloid formation. Furthermore, desynchronization of the suprachiasmatic nucleus (SCN) with related structures in the brain (15) as well as accumulation of tauopathy in the SCN (16) has been observed in AD or in animal models of AD. The SCN, located in the hypothalamus, is often described as the master-clock of the circadian system (17) which controls not only the secretion of melatonin and cortisol but also influences expression of clock genes such as Brain and muscle arnt-like protein 1 (BMAL1), CLOCK, PERIOD1, PERIOD2 (18). The circadian system not only steers wakefulness and sleep via the HPA axis but also influences locomotor activity and rest/activity patterns (19,20). Physical exercise on the contrary is itself able to influence and entrain the circadian system (21). Therefore, it can be speculated that a connection between altered activity patterns and disrupted circadian synchrony is present in individuals with incipient Alzheimer’s Disease. Of note, alterations of daily locomotor activity patterns has been observed in other forms of neurodegenerative diseases such as Parkinson’s disease (22,23) or Huntington’s disease (24). 

In addition to clarifying the motivation for the use case of MCI patients we now included a section on the hypothesized difference between MCI patients and healthy older volunteers as to possible mechanistic backgrounds. 

The introduction now reads as follows:

“In early stages of neuronal degeneration reduction of the suprachiasmatic nucleus (SCN) has been observed (17). The SCN has often been described as the central clock of the circadian network which controls melatonin and cortisol secretion and consequently influences alertness and sleep behavior (18, 19). Likewise a disruption of circadian rhythmicity, e.g., in the melatonin secretion cycle, has been demonstrated in patients with symptoms of neurodegenerative disorders (16, 19). Daily rest-activity patterns are hypothesized to be controlled by circadian functions, and its disruption has been observed in patients with AD and in mouse models of AD (20-24). Therefore, assessment of activity patterns might be well-suited to indicate AD pathology in observational studies.”

C#5: I also worried about dividing a sample of 13 patients with MCI by chronotype and then comparing this tiny group to any other group. It looks like fishing for significance.

R#5: We thank the editor for raising his concerns with respect to the above mentioned subgroup analysis. Our initial goal was to introduce the FoSR’s potential to incorporate covariates as in simple linear regression models, rather than to infer possible causal relationships. We do agree though that the small size of the subgroups within the different chronotypes of the MCI patients do not result in very robust estimates and might thus not be helpful for the reader. Thus, we decided to leave out all analyses on secondary endpoints from the manuscript and to concentrate on the differences between the two groups. To still address the FoSR’s high potential in analyzing such data we used age and sex as covariates to adjust for confounding. 

Please see respective changes to figure 1 of the manuscript:

Reviewer #1: 

General: The authors are applying a relatively novel analysis, function-on-scalar regression, to differentiate changes in diurnal activity between older adults with and without MCI. The authors note that this method is an improvement from aggregated approaches of determining links between activity and covariates such as demographic and health measures. Overall, this is a sound study and is utilizing an appropriate model to extend our knowledge of activity based measures in the presence or absence of disease in older adults.

C #1: The type of basis function used may improve or hurt prediction accuracy. You should compare model accuracy of different basis functions through either cross-validation for each type of basis function or inspection of the residual curves produced by the model and then determine which is the better fit.

R #1: We thank Mr. Hooyman for discussing the choice of basis functions used. All possible choices such as sin functions, cubic splines or wavelets have in common that they remove noise from the data since all act as low-pass filters. The reason why we chose wavelet basis functions is that they combine the advantages of sin functions and splines. Smoothing with wavelets and with sin functions give always periodic results. On the other hand, similar to cubic splines wavelets are localized in time (see fig. 1).

Figure 1: Six of the 18 Debauchies Wavelet basis function displayed over a 24 hrs time frame.

As a consequence, each wavelet basis function represents change of activity at a specific time of the day (e.g., the basis function at the bottom of fig. 1 represents change in activity between 10 p.m. and midnight). Formal testing of model accuracy is unfortunately in these settings not possible since the modeling of activity data by functions aims to smooth the data, meaning that of course the usage of more functions has a better model fit in modeling the data but at a price of a high rate of noise and a low rate of smoothing. In contrast a lower number of basis functions results in high rate of smoothing and low rate of noise but may smooth too much of the activity differences. Therefore we aimed to find a compromise between reducing noise but avoiding too much smoothing, that aligns with our research question (focused on longer phases of physical activity or inactivity over the day, rather than on very short intensive activities). In figure 3 different options with 9 or 36 basis functions are illustrated. Note that there is no possibility of choosing for example 20 or 10 functions. The next lower number of functions would be 9 resulting in highly smoothed curves that might not be sensitive to more subtle changes in activity over the day. The next higher number of functions would be 36 resulting in less smoothing, but would include too much noise that we are not interested in (see fig. 2). Using 18 basis functions each function represents 80 minutes of the day, when using 9 functions, each function would represent 160 minutes, when using 36 function, each would represent 40 minutes. We therefore decided to choose 18 basis functions to accurately model activity data in our patients. 

C #2: On line 206 you state that data should be aggregated into at least 5 minute epochs which is what you chose moving forward. What is the rationale for not choosing a smaller epoch of less than 5 minutes?

R #2: We thank Mr. Hooyman for giving us the opportunity to clarify our rationale for our epoch length. Our goal was to identify activity patterns which do not necessarily change within very short time periods. Imagine that standing at a traffic light might take up to one minute and would theoretically result in no acceleration while the person might be on a swift walk before and after the traffic light. Our aim was to detect differences in activity patterns over the course of a day, thus focusing on longer time frames with high / middle / low physical activity and not e.g. very short intensive physical activity times within longer time ranges of more or less inactivity. Previous studies have used different epoch lengths; we followed Xiao and colleagues (25) who used five minute epochs. Please note that Goldsmith and colleagues (1) used ten minute intervals which we thought would be too imprecise with respect to the beginning and end of activity. 

We clarified our rationale in the methods section which now reads as follows:

“We summarized the activity data using 5 minutes epochs resulting in 288 count data sets per participant per day (24h=1440 minutes=288*5 minutes)(10). The rationale for the 5 minutes epoch was that we aimed to identify group differences of activity levels over the course of a day, focusing on identifying longer periods with high/middle/low physical activity, respectively.”

C #3: On line 243 you state that 18 basis functions seemed to be a good compromise between smoothing out noise versus preserving activity patterns. Since there is little previous research to support this methodology the determination of number of basis functions should be driven by model comparison (i.e., comparison of residual curves produced by each model to detect bias in prediction as a result of number of basis functions). It is difficult to know if 18 basis functions is best without any tests of cross validation or other model comparison metrics.

R #3: The reviewer is correct that the choice of number of basis functions seems rather arbitrary, so we now detail on its underlying rationale: In figure 3 different options with 9 or 36 basis functions are illustrated. Note that there is no possibility of choosing for example 20 or 10 functions. The next lower number of functions would be 9 resulting in very smoothed curves that might not be sensitive to more subtle changes in activity over the day. The next higher number of functions would be 36 resulting in less smoothed curves that include still too much noise that we are not interested in. (see fig. 2). Using 18 basis functions each function represents 80 minutes of the day, when using 9 functions, each function would represent 160 minutes, when using 36 function, each would represent 40 minutes. 

Figure 2: Six of the 36 Debauchies Wavelet basis function displayed over a 24 hrs time frame.

Given our hypothesis that the circadian rhythm of MCI patients would be disturbed, with higher activity during night and lower activity in the morning hours compared to healthy older controls, we wanted to examine activity for longer time-intervals than one hour only. FoS regression with k=18 basis functions would be well suited to detect such differences. Moreover, k=18 basis functions would provide acceptable smoothing without suppressing important changes in activity that occur at a scale longer than two hours (see fig. 3).

Figure 3: Smoothing of a activity data from one sample subject with 36 (blue line), 18 (red line) or 9 (black dotted line) Debauchies Wavelet basis functions.

C #4: What are the results in table 1 reported with an unadjusted p-value? Isn’t each row of this table an independent test and thus inflating your chance of a type 1 error?

R #4: Thank you for giving us the opportunity to correct our captions for table 1. In the beginning we tested for group differences but found the issue of inflating type 1 error misleading. We therefore decided to report standardized mean differences as effect measures to enable the reader to understand the magnitude of effects between the two groups. We now deleted the caption in which we reported differences with a p-value smaller than 0.05. 

C #5: In reviewing the cited Goldsmith paper, https://www.ncbi.nlm.nih.gov/pmc/articles/PMC4987214/, shouldn’t the Function-of-Scalar-Regression (FoSR) give you a global coefficient and p-value for each covariate used in your model? The stated purpose of this paper should be to compare results of multiple linear regression using summer activity levels compare to results of the FoSR in terms of global beta and p-values, just like that referenced Goldsmith paper. See table 1 from the linked study above. This would fulfill the primary purpose of this paper to demonstrate the sensitivity of the FoSR analysis compared to more traditional regression approaches.

R #5: We thank Mr Hooyman for this important point and for highlighting the relation to the original method from Goldsmith et al. The assumption of a global p-value is that the activity between the two groups differ at any given time. We performed a global test under the null hypothesis of beta(t) = 0 for any t = 0, …, 287 and received a p-value of < 0.0001. Therefore we can conclude that beta(t) is smaller or larger than zero for any time point between 0 and 287. That is a statement on a global level with little informative value. As the relevance for the general audience of such a statement is low we decided to not report a global p-value. Of note, it is also not possible to derive an effect size on a global coefficient. Therefore, it is not possible to interpret the magnitude on e.g. global group differences.

C #6: Also, does your FOSR model only include covariates of group (MCI vs HOV) and age? In line 279 you state that you only control for age to prevent overfitting but how do you know that you have overfit the model in this context without doing some level of cross validation which is absent here?

R #6: Overfitting of FoSR models is not as straight forward as in simple linear regression as also pointed out by Goldsmith and colleagues:

“Although degrees of freedom are less easily defined for FoSR models, a useful rule of thumb is to compare the number of subjects to the number of coefficient functions as a guide for decisions about model size and complexity.” (1)

Please be aware that Goldsmith and colleagues had a sample of 420 children. Therefore, they were able to include several covariates in one model. Our sample comprises only 66 subjects. We therefore decided to include only age as covariate. Further, we now added sex as a covariate as we did not perform any subgroup analyses using the MEQ as covariate. 

We clarified the rationale of overfitting our data and rephrased the respective statement as follows:

“All analyses were only controlled for age and sex in order not to overfit the model following the suggestions of Xiao et al. that the number of subjects included in the model should steer the decision on model size and complexity. (11)”

C #7: 

And again in reference to the previous Goldsmith paper when you are plotting relative activity in figures 1 or 2 is this similar to their plotting of the coefficient function (i.e. the change in the prediction of activity level as a function of the covariate across time)? See figure 1 and 2 in Goldsmith 2016. I continue to bring up the Goldsmith paper because it would be in the best interest of this paper to demonstrate the strength of this method based on how it has been used previously to help identify differences and similarities in your approach and interpretation compared to how it was used previously.

R #7: In our manuscript we decided not to plot coefficient functions, as they are more difficult to interpret and show very similar information as the relative activity graphs displayed in our manuscript. By delogarithmizing the coefficient function we believe the reader will more easily understand the meaning of the differences displayed in the figures and thus acknowledge the value of the FoSR as a statistical tool for clinical research.

C #8: In figure 1 you should demonstrate how the predicted values of the FoSR model differ from a simple averaging of the data in both the absolute and relative activity counts between groups. This would provide more face validity on the conceptual basis of how FoSR captures activity patterns in this sample of data that averaging and summing methods do not.

R #8: In line with the reviewer’s request, we included average activity counts in table 1 for both groups and calculated standardized mean difference. Also, we implemented the average activity count as horizontal lines in figure 1 A.

Table 1:

Physical activity HOV

(n = 48) MCI

(n = 18) SMD standardized mean difference

Average activity count per day, mean (SD) 426 (139) 396 (169) 0.19

Figure 1 A:

Distribution of a) mean absolute activity between MCI and HOV over time and b) mean relative activity between MCI and HOV over time with 95% Confidence Interval (CI) adjusted for age. Horizontal lines in Figure 1a) displays the average daily activity of log-transformed data from fitted curves of respective groups. The black line in Figure 1b) displays a relative activity. A value of one indicates no difference of activity between the two groups.

C #9: In the chronotype analysis are you still using FoSR to determine differences in activity level as a function of strata and group? It does not explicitly say if this is the case. I believe you say it on line 404 but it should be stated earlier.

R #9: With respect to the concerns raised by Dr. Oster we decided to leave out analyses on chronotypes in the revised version of our manuscript. In the initial version of our manuscript we did use FoSR also for the analyses on chronotypes.

C #10:You may also wish to include the experiment that each individual comes from as a random variable/effect which may also help control for any variance in this study since this sample is a pooled group from two previous projects.

R #10: Random effects are an important issue in the analysis of repeated measures and can easily be incorporated within the FoSR model approach. We implemented a random intercept term for subjects into the model. Please see respective sentence starting in line 192:

“In addition to the fixed effects, the model equation contains a random intercept to account for clustering of data within individuals.”

And in line 249:

“We included a random intercept in the model equation since some participants contributed several correlated 24 hours activity records to the study.”

As the original studies did not observe the effect of an intervention on activity patterns it does not seem reasonable to use the two individual studies as an additional random effect. Of note, the protocols as well as inclusion and exclusion criteria of the two pooled studies did not differ with respect to the data analyzed here. 

C #11: Line 258 do you mean that you inputted multiple 24-hour activity records from one individual into your model?

R #11: This is a very important issue also raised by reviewer #3. Thus, we are thankful for the opportunity to clarify the procedure and to highlight the advantages of the FoSR model. We used all available full days. In order to not lose the information on variance between single observation days we used a mixed model with random intercept.

C #12: Line 276 change epoque to epoch

R #12: We corrected the mistake.

C #13: Line 309 should OLD be HOV?

R #13: We corrected the mistake.

C #14: Line 335 change hours to hour

R #14: We corrected the mistake.

C #15: In figure caption for Figure 1 include what the black line in B represents.

R #15: We added the following sentence to the caption:

“The black line in Figure 1b) displays a relative activity. A value of one indicates no difference of activity between the two groups.”

Reviewer #2: 

C #1:The data and question could be used to write a useful paper but this one tries to do too much with too little and isn't careful about how it is done. The motivation is not appropriate for the question, the paper contains irrelevant material, the analytic sample is tiny and extremely selected, so I place no confidence in the results.

R #1: We thank Reviewer #2 for addressing relevant concerns. We have rephrased the introduction section to render the motivation for the manuscript clearer to the reader, increased the sample size through reevaluation of previously excluded data, left out any subgroup analyses and left out any non-relevant endpoints. Here, we would highlight the key points of our revised manuscript:

- Demonstration of a model-based analysis of time-based accelerometer data

- Use case in a well-defined and clinically relevant sample

- Statistical model that is capable of testing differences over a 24-hours cycle while adjusting for multiple comparison

- Capability of adjusting for confounding factors

We hope that we are now able to convey the motivation for the manuscript. Accordingly, we would like to refer also to the responses given to the comments raised by the other two reviewers and by the editor.

Reviewer #3: 

Rackoll and colleagues describe the results of a “… time series analyses on continuous accelerometry data – a clinical example in older adults with and without cognitive impairment” and find that patients with mild cognitive impairment show an elevation in activity levels, specifically in the morning and the afternoon.

C #1: The first sentence of the MS is: “Current analysis approaches of accelerometry data use sum score measures which do not provide insight in activity patterns over 24 hours, and thus do not adequately depict circadian activity patterns”.

What are the authors talking about? What is meant by “Current analysis approaches”? Do the authors refer to commercial applications that come with the accelerometers, or the read-outs of smart watches?

R #1: We thank Reviewer #3 to further allow us to clarify this misunderstanding. In that first sentence we aimed to state that most clinical research using accelerometer data aggregate amount of activity over the recording period and report some sort of sum measures such as percentage of time spent in different intensities of activity (26,27). We assume that this approach is also motivated by software provided by manufacturers of accelerometers used in research. Actigraph corp. for example uses their own software Actilife which aggregates activity counts averaged over all recording days (for further information please see: https://actigraphcorp.com/actilife/). Apart from functionalities provided by software, we think that reporting of sum scores do reflect current reporting methods in clinical research (28–30). Furthermore, we stated that these measures are not detailed enough to analyze activity data given the assumption that the data is seen as a time-series. Sum measures neglect the detailed nature of time-based data and therefore only reflect a fraction of information entailed in accelerometer data.

According to this comment we rephrased the first sentence of the abstract which now reads as follows:

“Many clinical studies reporting accelerometry data use sum score measures such as percentage of time spent in moderate to vigorous activity which do not provide insight into differences in activity patterns over 24 hours, and thus do not adequately depict circadian activity patterns.” 

C #2: Whatever is meant by this introductory sentence, it is far from reality and shows that the authors are not familiar with the field of circadian research. Time-series analysis of circadian data has been uses decades before even accelerometers were applied in human studies, for example by Sokolove in the 1970s. But even today and specifically concerning actimetry in humans, the search – “time series analysis" circadian human actimetry – yields more than 350 papers, 16 alone in 2020. The top hit in relevance is a paper called “Multiscale adaptive analysis of circadian rhythms and intradaily variability: application to actigraphy time series in acute insomnia subjects”, which even has been published in PLOS one. I am willing to bet that all of these papers go beyond “sum score measures”. The authors should look at this paper as an example of what can be done when analysing “continuous accelerometry data”.

R #2: We thank Reviewer #3 to give us the chance to clarify the motivation of applying time series analyses on accelerometry data of a clinical population. In our manuscript we aimed to provide arguments for researchers working with clinical populations to more broadly apply Function-on-Scalar-regression and to highlight the advantages of this particular method for time-based data. Of note, usage of time series analyses has been identified to be used only rarely in accelerometer measures of physical activity (27). We acknowledge the vast field of functional data analysis or time-series analysis – but it is not the standard in medical literature to apply such methods on accelerometer data. Here, we are interested in activity profiles and its inherent patient specific identifiers to further support early diagnosis or identification of patients at risk – as differences between populations will only be subtle at an early stage of cognitive decline. Furthermore, FoSR is a promising method which can be applied to any time-based or functional periodic data. We also want to highlight with respect to the field of data transformation, Debauchier wavelets in particular are proposed as a promising method for time series analyses but are also not used on a regular basis (31). In addition to the revised introduction which we now believe better communicates the motivation of our study, we would like to take the chance to discuss the advantages of our method in comparison to the study “Multiscale adaptive analysis of circadian rhythms and intradaily variability: application to actigraphy time series in acute insomnia subjects” by Fossion and colleagues mentioned in comment #2 above. In this study two different methods were compared to identify circadian rhythms in the rest-activity profiles of young insomnia patients compared to young healthy controls. Fossion et al discuss in a very detailed and stringent way two different methodological approaches to assess circadian rhythms. In this study singular spectrum analysis (SSA) is used in comparison with traditional cosinor analyses and measures of intraday variability. SSA is proposed as a more advanced method as it better reflects the inherent circadian and ultradian rhythms as well as the trend of the data. Here, our proposed method is not superior to the SSA but has some advantages that we argue to be very relevant in clinical research. First, FoSR is a model-based approach. Statistical analyses can be adjusted for confounding factors – a very important issue in the light of a somewhat heterogeneous clinical population. Additionally, this is also relevant if prognostic factors are to be evaluated. We demonstrated the potential of the model-based FoSR approach by adjusting for age and sex and could show group differences in activity pattern even after adjustment for age which itself has been observed to be associated to circadian rhythms (32). 

In the above mentioned manuscript the results of the SSA are similar to a cosinor analysis with the advantage of an integration of intraday variability. Since data from several days of the same volunteer may be correlated, we extended the FoSR algorithm described in the original publication from Goldsmith et al. 2016 to address individual intraday variability. In our approach we used a mixed model (with random intercept) instead of an ordinary linear model as in Goldsmith's article. The results of the FoSR presented in our manuscript are also more intuitive and easier to interpret for a reader not familiar with circadian rhythm analysis. The differences in activity patterns at a specific time point are displayed, instead of showing changes of acrophases or periods which need further interpretation. Of note, it is also a straightforward approach in the FoSR model to correct for multiple comparison which is also relevant for measures such as the SSA but has to be done as a second step there.

To clearly communicate the motivation for our manuscript we rephrased the introduction section in the abstract which now reads as follows:

“Here, we present an improved functional data analysis approach to model activity patterns and circadian rhythms from accelerometer data. As a use case, we demonstrated its application in patients with mild cognitive impairment (MCI) and age-matched healthy older volunteers (HOV).“

In addition, we have rewritten the aim of our study in the introduction section of the manuscript which now reads:

“Here, as a clinical example we applied a functional data analysis algorithm on healthy older volunteers (HOV) and patients with mild cognitive impairment (MCI) in a previously acquired data set to describe the circadian rhythm in both groups, and to test if their activity patterns differed throughout a 24 hours cycle. We hypothesize that our algorithm would be able to detect differences in distribution of activity with regard to timing and magnitude between HOV and MCI. Furthermore, we provide with detailed methodology in the supplemental appendix, and our analysis scripts are available online (see https://doi.org/10.5281/zenodo.3718578) to encourage future applications of this approach.“

 C #3: It seems that the authors use FoSR similar to how some people use SPSS, i.e., without actually looking at the data. The figures the authors come up with are simple 24-hour averages as have been used in analyses of human actimetry for decades. It is simply not enough to state that there are activity level-differences between the groups. Actimetry can do so much more, can dig so much deeper than the analysis presented here. The first step of every detailed analysis of circadian long-term time series are making double plots, so that one can see how the activity is generally spread over every day’s 24 hours. The PLOS one paper mentioned above, goes into “intradaily variability”, which could well be at the basis of the activity level differences. We have no idea how dispersed or consolidated the daily activity profiles are in the participants of the two groups, we have no idea when the participants apparently sleep (which could be assessed by looking at the data).

R #3: We like to clarify some aspects of our methodology, which were questioned by Reviewer #3. In order to account for intraday variability we implemented a random intercept in our model, which is an extension to the original method by Goldsmith and colleagues. It takes into account the irregular number of continuously recorded days, which might be a problem with especially elderly or cognitive impaired participants. We would also like to reject the criticism that we did not adequately look at our data. During analysis all periods of sleep were assessed manually by two assessors and these times are reported in table 1. Details were given in the supplemental appendix. To provide more insight into the methodology we inserted the sleep scoring section from the appendix into the method section of the manuscript and included a plot with the raw values from two sample participants.

Respective changes in our methods section now reads as follows

“Sleep scoring is based on the amount of activity counts per epoch and on the directionality of the sensors. Sleep onset and awakening time were rated visually by observation of a sharp decrease or increase of signal by two independent and experienced raters and controlled with sleep diaries administered to the patients before accelerometry assessment (32). For automated sleep scoring we used the Cole-Kripke algorithm within the ActiLife software. The software then automatically calculates total time in bed, total sleep time, sleep latency, number of arousals, average duration of awakening and wake after sleep onset.”

New figure 2 looks as follows:

Distribution of absolute activity over a five day period for A) one HOV sample participant and B) one MCI sample participant. Red lines denote the wavelet smoothing. 

Regarding the use of the random intercept please see respective sentence starting in line 192:

“In addition to the fixed effects, the model equation contains a random intercept to account for clustering of data within individuals.”

And in line 249:

“We included a random intercept in the model equation since some participants contributed several correlated 24 hours activity records to the study.”

C #4: Figure 1a should be also double-plotted, since this allows a proper representation of sleep. Even a relative short recording of a week allows to determine a mid-trough time of something like time of “L5” (often used in time series analysis of human actigraphy and mentioned in the present MS). These individual phases would have been much more appropriate to be used in a normalisation process compared to the MEQ questionnaire. It is also not clear whether individual daily profiles were normalised to individual mean levels (deviation from daily means or entered moving means). If one wants to make statutes about specific times of day where activity levels differ between groups, such a step would be essential.

R #4: Here we would like to clarify our methodology. Accelerometer data is in itself quite noisy as accelerations in a Cartesian space are not constant. Therefore some averaging or smoothing is necessary. Instead of applying a method such as moving average we decided to use a discrete Debauchies Wavelet smoothing as this enables us to take into account the entire 24-hour cycle as well as considering local means. Furthermore, we agree with Dr. Oster’s comment that our sample is too small to differentially model the effect of chronotypes on activity distributions. We therefore left out the Morningness-Eveningness Questionnaire as a covariate in our model. We argue though that in larger samples controlling for chronotype might be relevant to more clearly describe activity differences in neuropathological cohorts.

C #5: the only result that came out of this “novel” FoSR analysis is the vague activity level difference that can be sen in Fig 1a, then the field does not need this analysis. This is a shame because we need much more good analysis of actimetry data from every kind of patients, especially in psychiatry and for higher age-groups.

I suggest the authors team up with people, who have long-term experience in circadian analysis of activity recordings and milk their data again with all the insights available for this type of analysis.

R# 5: We hope that we were able to convince Reviewer # 3 of the value of our manuscript for the medical field and that we were able to give better insight on the motivation of our study.

References

1. Goldsmith J, Liu X, Jacobson JS, Rundle A. New Insights into Activity Patterns in Children, Found Using Functional Data Analyses. Med Sci Sports Exerc. 2016;48(9):1723–9. 

2. Choi L, Liu Z, Matthews, Charles E, Buchowski MS. Validation of Accelerometer Wear and Nonwear Time Classification Algorithm. Med Sci Sport Exerc. 2011;43(2):357–64. 

3. Ae Lee J, Gill J. Missing value imputation for physical activity data measured by accelerometer. Stat Methods Med Res. 2018;27(2):490–506. 

4. National Institute for Health and Care Excellence. Dementia, disability and frailty in later life-mid-life approaches to delay or prevent onset NICE guideline [Internet]. 2015 [cited 2020 Nov 17]. Available from: www.nice.org.uk/guidance/ng16

5. Brasure M, Desai P, Davila H, Nelson VA, Calvert C, Jutkowitz E, et al. Physical activity interventions in preventing cognitive decline and alzheimer-type dementia a systematic review [Internet]. Vol. 168, Annals of Internal Medicine. American College of Physicians; 2018 [cited 2020 Sep 23]. p. 30–8. Available from: https://pubmed.ncbi.nlm.nih.gov/29255839/

6. Bahar-Fuchs A, Martyr A, Goh AMY, Sabates J, Clare L. Cognitive training for people with mild to moderate dementia [Internet]. Vol. 2019, Cochrane Database of Systematic Reviews. John Wiley and Sons Ltd; 2019 [cited 2020 Sep 23]. Available from: https://pubmed.ncbi.nlm.nih.gov/30909318/

7. Mcgrattan AM, McEvoy CT, McGuinness B, McKinley MC, Woodside J V. Effect of dietary interventions in mild cognitive impairment: A systematic review [Internet]. Vol. 120, British Journal of Nutrition. Cambridge University Press; 2018 [cited 2020 Sep 23]. p. 1388–405. Available from: /pmc/articles/PMC6679717/?report=abstract

8. Rock PL, Roiser JP, Riedel WJ, Blackwell AD. Cognitive impairment in depression: A systematic review and meta-analysis [Internet]. Vol. 44, Psychological Medicine. Cambridge University Press; 2014 [cited 2020 Sep 23]. p. 2029–40. Available from: https://pubmed.ncbi.nlm.nih.gov/24168753/

9. Yaffe K, Falvey CM, Hoang T. Connections between sleep and cognition in older adults. Lancet Neurol. 2014;13(10):1017–28. 

10. Cross N, Terpening Z, Rogers NL, Duffy SL, Hickie IB, Lewis SJG, et al. Napping in older people “at risk” of dementia: Relationships with depression, cognition, medical burden and sleep quality. J Sleep Res. 2015;24(5):494–502. 

11. Satlin A, Volicer L, Stopa EG, Harper D. Circadian locomotor activity and core-body temperature rhythms in Alzheimer’s disease. Neurobiol Aging. 1995;16(5):765–71. 

12. Yesavage JA, Friedman L, Kraemer HC, Noda A, Wicks D, Bliwise DL, et al. A follow-up study of actigraphic measures in home-residing Alzheimer’s disease patients. J Geriatr Psychiatry Neurol [Internet]. 1998;11(1):7–10. Available from: isi:000074755600002

13. Cochrane A, Robertson IH, Coogan AN. Association between circadian rhythms, sleep and cognitive impairment in healthy older adults: An actigraphic study. J Neural Transm. 2012 Oct 10;119(10):1233–9. 

14. Oyegbami O, Collins HM, Pardon M-C, Ebling FJP, Heery DM, Moran PM. Abnormal Clock Gene Expression and Locomotor Activity Rhythms in Two Month-Old Female APPSwe/PS1dE9 Mice. Curr Alzheimer Res. 2017;14(8):850–60. 

15. Cermakian N, Waddington Lamont E, Boudreau P, Boivin DB. Circadian clock gene expression in brain regions of Alzheimer’s disease patients and control subjects. J Biol Rhythms. 2011;26(2):160–70. 

16. Stevanovic K, Yunus A, Joly-Amado A, Gordon M, Morgan D, Gulick D, et al. Disruption of normal circadian clock function in a mouse model of tauopathy. Exp Neurol. 2017;294:58–67. 

17. Coogan AN, Schutová B, Husung S, Furczyk K, Baune BT, Kropp P, et al. The circadian system in Alzheimer’s disease: Disturbances, mechanisms, and opportunities. Biol Psychiatry. 2013;74(5):333–9. 

18. Zelinski EL, Deibel SH, McDonald RJ. The trouble with circadian clock dysfunction: Multiple deleterious effects on the brain and body. Neurosci Biobehav Rev. 2014;40:80–101. 

19. Lim ASP, Yu L, Costa MD, Leurgans SE, Buchman AS, Bennett DA, et al. Increased fragmentation of rest-activity patterns is associated with a characteristic pattern of cognitive impairment in older individuals. Sleep. 2012;35(5):633-40B. 

20. Musiek ES, Bhimasani M, Zangrilli MA, Morris JC, Holtzman DM, Ju Y-ES. Circadian Rest-Activity Pattern Changes in Aging and Preclinical Alzheimer Disease. JAMA Neurol. 2018 May 1;75(5):582–90. 

21. Tahara Y, Aoyama S, Shibata S. The mammalian circadian clock and its entrainment by stress and exercise. Vol. 67, Journal of Physiological Sciences. Springer Japan; 2017. p. 1–10. 

22. Willis GL, Kelly AMA, Kennedy GA. Compromised circadian function in Parkinson’s disease: Enucleation augments disease severity in the unilateral model. Behav Brain Res. 2008 Nov 3;193(1):37–47. 

23. Leng Y, Blackwell T, Cawthon PM, Ancoli-Israel S, Stone KL, Yaffe K. Association of Circadian Abnormalities in Older Adults with an Increased Risk of Developing Parkinson Disease. JAMA Neurol. 2020;77(10):1270–8. 

24. Morton AJ, Wood NI, Hastings MH, Hurelbrink C, Barker RA, Maywood ES. Disintegration of the Sleep-Wake Cycle and Circadian Timing in Huntington’s Disease. 2005; 

25. Xiao L, Huang L, Schrack JA, Ferrucci L, Zipunnikov V, Crainiceanu CM. Quantifying the lifetime circadian rhythm of physical activity: A covariate-dependent functional approach. Biostatistics. 2015;16(2):352–67. 

26. Shiroma EJ, Schrack JA, Harris TB. Accelerating Accelerometer Research in Aging. Journals Gerontol Ser A [Internet]. 2018 Apr 17 [cited 2018 Oct 26];73(5):619–21. Available from: https://academic.oup.com/biomedgerontology/article/73/5/619/4955526

27. Silva SSM, Jayawardana MW, Meyer D. Statistical methods to model and evaluate physical activity programs, using step counts: A systematic review. PLoS One. 2018;13(11):1–19. 

28. Lim SER, Ibrahim K, Sayer AA, Roberts HC. Assessment of Physical Activity of Hospitalised Older Adults: A Systematic Review. J Nutr Heal Aging. 2018;22(3):377–86. 

29. Bassett DR, Toth LP, LaMunion SR, Crouter SE. Step Counting: A Review of Measurement Considerations and Health-Related Applications. Sport Med. 2017;47(7):1303–15. 

30. Migueles JH, Cadenas-Sanchez C, Ekelund U, Delisle Nyström C, Mora-Gonzalez J, Löf M, et al. Accelerometer Data Collection and Processing Criteria to Assess Physical Activity and Other Outcomes: A Systematic Review and Practical Considerations. Sport Med. 2017;47(9):1821–45. 

31. Leise TL. Analysis of Nonstationary Time Series for Biological Rhythms Research. J Biol Rhythms. 2017;32(3):187–94. 

32. Huang Y, Liu R, Wang Q, Someren EJW Van, Xu H, Zhou J. Age-associated difference in circadian sleep – wake and rest – activity rhythms. Physiol Behav. 2002;76:597–603.

---

## [Decision Letter · Decision Letter 1]

4 Jan 2021

PONE-D-20-08099R1

Applying time series analyses on continuous accelerometry data – a clinical example in older adults with and without cognitive impairment

PLOS ONE

Dear Dr. Rackoll,

Thank you for submitting your manuscript to PLOS ONE. After careful consideration, we feel that it has merit but does not fully meet PLOS ONE’s publication criteria as it currently stands. Therefore, we invite you to submit a revised version of the manuscript that addresses the points raised during the review process.

Please carefully address the concerns raised by the reviewer.

We look forward to receiving your revised manuscript.

Kind regards,

Henrik Oster, Ph.D.

Academic Editor

PLOS ONE

Reviewers' comments:

Reviewer's Responses to Questions

**Comments to the Author**

1. If the authors have adequately addressed your comments raised in a previous round of review and you feel that this manuscript is now acceptable for publication, you may indicate that here to bypass the “Comments to the Author” section, enter your conflict of interest statement in the “Confidential to Editor” section, and submit your "Accept" recommendation.

Reviewer #1: (No Response)

2. Is the manuscript technically sound, and do the data support the conclusions?

Reviewer #1: Partly

3. Has the statistical analysis been performed appropriately and rigorously? 

Reviewer #1: No

4. Have the authors made all data underlying the findings in their manuscript fully available?

Reviewer #1: Yes

5. Is the manuscript presented in an intelligible fashion and written in standard English?

Reviewer #1: Yes

6. Review Comments to the Author

Reviewer #1: The purpose of this paper is to use Function on Scalar Regression to determine if activity levels between healthy older volunteers and individuals with Mild Cognitive Impairment throughout different times of the day.

I believe the authors have done a good job of addressing my initial comments. However, with further inspection of the cited research on FOSR and code provided new concerns are presented.

In my second comment I asked the reviewers to provide a rationale for why they use a 5-minute epoch which they have now provided. Looking at the original Goldsmith paper (11) their rationale for epoch length is not based on activity patterns but rather achieving a normal distribution of count data at each time point. This comment by Goldsmith represents an important statistical consideration and the authors should confirm the same is true for their sample given the smaller sample size used here compared to Goldsmith (11). Unintended outliers that are generated at certain timepoints due to a selected epoch length may skew results of the FOSR analysis. You state that you log transform the data to correct for skew but that may be an unnecessary step if the right epoch is chosen. Also, you state in response to comment 1 that your focus is on longer phases of activity over the day, yet you choose a shorter epoch than previous studies see Goldsmith (11).

In response to comment 3 you state your design of your wavelet function is based on your hypothesis that MCI patients would have higher activity at night and lower during the morning compared to HOV. This is not your hypothesis in the manuscript, however. On line 110 you make a more general hypothesis that your algorithm will detect differences in activity in both timing and magnitude between the two groups. No where in the introduction do you state any evidence on how circadian rhythms would be disrupted in terms of specific time of day. If there is evidence it needs to be stated. Your hypothesis would also need to be updated in the manuscript. It is wearisome that you use one hypothesis to justify a methodological decision in the comments but then state a different hypothesis in the manuscript.

More to the point, your rationale for using a wavelet function is based on research that uses this type of function on time series data but not necessarily research that has used FOSR. Goldsmith (11) provides all the code necessary to use FOSR as it was applied in their paper which seems to be the fundamental goal of this paper, i.e. use FOSR to determine differences in activity patterns between MCI and HOV volunteers. However, the Goldsmith R package, refund, does not include a function to use wavelets for smoothing data, which is why I believe you go this route of using a linear mixed effects model instead. Ultimately, this smoothing step is of huge importance in the application of FOSR where the level of smoothing may produce differences between groups that do not necessarily exist. You justify the use of the wavelet function based on previous research using it on time series data but has it ever been used on a geriatric population? Or have the number of basis functions been validated to represent true activity versus noise? Obviously, a wavelet function with X number of basis functions may be more accurate for one group or individual than another, for example, 36 basis functions for kids versus 9 for older adults. If you have no direct evidence validating the application of wavelets with a specific number of basis functions to the activity levels of the groups investigated here, I would recommend dropping this smoothing step altogether. Alternatively, I recommend that the authors utilize the Goldsmith package, refund, to implement FOSR as it has been used in the past, thus making the methods of this paper more directly comparable to the research they claim to base the current results off of. Additionally, the use of the refund package would allow for more direct model comparison between models that include or exclude the covariate of group while controlling for age and sex. Code to do all this can be found here: https://jeffgoldsmith.com/papers.html. CRAN for refund here: https://cran.r-project.org/web/packages/refund/refund.pdf.

The number of basis functions used in your wavelet transformation also dictates your capability to determine statistical significance at different times based on your use of a bonferonni correction to account for multiple comparisons. Why not use a false discovery rate correction instead? Applying Bonferroni based on number of basis functions seems to make it either too conservative or too liberal and has a direct impact on your capability to determine when the activity patterns differ between groups. Also, the refund package uses an FOSR function that is Bayesian and therefore utilizes confidence intervals to make determinations of significance rather than p-values.

The authors also never run a multiple linear regression on the aggregate activity data. In the introduction the authors state that this method is in effectual but they never perform it to actually demonstrate that this is the case.

MINOR COMMENTS

Change hours to hour, line 110.

Be sure to cite the NHANES cohort, line 174.

Please cite who is suggesting that the use of cubic splines or wavelet transformations is suggested for actigraph data, line 206.

7. PLOS authors have the option to publish the peer review history of their article (what does this mean?). If published, this will include your full peer review and any attached files.

Reviewer #1: **Yes: **Dr. Andrew Hooyman

---

## [Author Response · Author response to Decision Letter 1]

12 Apr 2021

Reviewer #1: 

The purpose of this paper is to use Function on Scalar Regression to determine if activity levels between healthy older volunteers and individuals with Mild Cognitive Impairment throughout different times of the day.

I believe the authors have done a good job of addressing my initial comments. However, with further inspection of the cited research on FOSR and code provided new concerns are presented.

Comment #1:

In my second comment I asked the reviewers to provide a rationale for why they use a 5-minute epoch which they have now provided. Looking at the original Goldsmith paper (11) their rationale for epoch length is not based on activity patterns but rather achieving a normal distribution of count data at each time point. This comment by Goldsmith represents an important statistical consideration and the authors should confirm the same is true for their sample given the smaller sample size used here compared to Goldsmith (11). Unintended outliers that are generated at certain timepoints due to a selected epoch length may skew results of the FOSR analysis. You state that you log transform the data to correct for skew but that may be an unnecessary step if the right epoch is chosen. Also, you state in response to comment 1 that your focus is on longer phases of activity over the day, yet you choose a shorter epoch than previous studies see Goldsmith (11).

We thank Reviewer #1 for allowing us to further clarify how our approach (rationale given in the previous version) relates to Goldsmith et al. 

First, we concede that pre-investigating the distribution of the data is important to minimize risks of bias due to outliers. However, an approach that chooses epoch lengths with the aim to achieve normally distributed data might lead to impractical epoch length (e.g. seven minutes could be the smallest epoch length with normally distributed data but a 24 hour cycle cannot be divided into seven minute epochs). Thus, our decision on epoch length was not driven by the aim to achieve a normal distribution of count data at each time point. We decided to define an epoch length on conceptual grounds in which we aimed “to detect differences in activity patterns over the course of a day” (for details see our response from the last revision) and to subsequently use data transformations that avoid skewness. Log-transformation is a simple data transforming step that is easy to apply, while the chances of introducing bias are limited, rendering it an appropriate and economic strategy for other clinical researchers (Bland & Altman, BMJ, 1996). Of note, also Xiao and other authors from the ‘refund’ package log-transformed their count data due to skewness (Xiao et al, Biostatistics, 2015). 

Second, our comment from the previous revision (“focusing on identifying longer periods”) was related to the alternative of using very short epochs such as one minute. We revised the wording for our rationale in the Methods section from our previous revision by deleting the word ‘longer’. The respective sentence now reads as follows:

“We summarized the activity data using 5 minutes epochs resulting in 288 count data sets per participant per day (24h=1440 minutes=288*5 minutes)(10). The rationale for the 5 minutes epoch was that we aimed to identify group differences of activity levels over the course of a day, focusing on identifying periods with high/middle/low physical activity, respectively.”

Comment #2:

In response to comment 3 you state your design of your wavelet function is based on your hypothesis that MCI patients would have higher activity at night and lower during the morning compared to HOV. This is not your hypothesis in the manuscript, however. On line 110 you make a more general hypothesis that your algorithm will detect differences in activity in both timing and magnitude between the two groups. No where in the introduction do you state any evidence on how circadian rhythms would be disrupted in terms of specific time of day. If there is evidence it needs to be stated. Your hypothesis would also need to be updated in the manuscript. It is wearisome that you use one hypothesis to justify a methodological decision in the comments but then state a different hypothesis in the manuscript.

We want to use the opportunity to clarify a misunderstanding from comments given in our last revision. In our comment #3 from our last response, we compared 40 minute intervals (36 basis functions) with 80 minute (18 basis functions) and 120 minute intervals (9 basis function). Our choice of the number of basis functions was based on our hypothesis that we would thus detect group difference that are evenly distributed over the course of a day. We considered an underlying time window that was neither too small nor too large. In our statement in comment #3 we wanted to give an example of what time distributions we were aiming at. Within that sentence we accidently left out the word ‘for example’. It should have stated: “Given our hypothesis that the circadian rhythm of MCI patients would be disturbed, with, for example, higher activity during night and lower activity in the morning hours compared to healthy older controls, we wanted to examine activity for longer time-intervals than one hour only.” Previous evidence indicates that differences in activity patterns are differentially spread over the day given circadian disruptions observed in patients with Alzheimer’s disease (e.g., increased wake after sleep onset and delayed wake times, Leng et al. Lancet Neurol, 2020; Musiek et al. Jama Neurol, 2018). There is not enough evidence on rest-activity cycles in MCI patients and also no strong indication how that influences activity patterns. Therefore, we would like to stay with the hypothesis provided in the manuscript and apologize for given a misleading explanation in the last revision. Data are not sufficient to ground a hypothesis on specific periods in which activity differs between MCI and HOV. 

Comment #3:

More to the point, your rationale for using a wavelet function is based on research that uses this type of function on time series data but not necessarily research that has used FOSR. Goldsmith (11) provides all the code necessary to use FOSR as it was applied in their paper which seems to be the fundamental goal of this paper, i.e. use FOSR to determine differences in activity patterns between MCI and HOV volunteers. However, the Goldsmith R package, refund, does not include a function to use wavelets for smoothing data, which is why I believe you go this route of using a linear mixed effects model instead. Ultimately, this smoothing step is of huge importance in the application of FOSR where the level of smoothing may produce differences between groups that do not necessarily exist. You justify the use of the wavelet function based on previous research using it on time series data but has it ever been used on a geriatric population? Or have the number of basis functions been validated to represent true activity versus noise? Obviously, a wavelet function with X number of basis functions may be more accurate for one group or individual than another, for example, 36 basis functions for kids versus 9 for older adults. If you have no direct evidence validating the application of wavelets with a specific number of basis functions to the activity levels of the groups investigated here, I would recommend dropping this smoothing step altogether. Alternatively, I recommend that the authors utilize the Goldsmith package, refund, to implement FOSR as it has been used in the past, thus making the methods of this paper more directly comparable to the research they claim to base the current results off of. Additionally, the use of the refund package would allow for more direct model comparison between models that include or exclude the covariate of group while controlling for age and sex. Code to do all this can be found here: https://jeffgoldsmith.com/papers.html. CRAN for refund here: https://cran.r-project.org/web/packages/refund/refund.pdf.

The reviewer addresses a valid point with respect to methodological aspects of the statistical approach. First, we want to highlight that the aim of our manuscript is to introduce clinical researchers to the advantages of FoSR for the assessment of accelerometer-based time series activity data. Although further research is needed on different aspects of the FoSR model statistics, validating the approach is not the goal of our current study and might obscure the manuscript’s agenda for the clinical reader. Reviewer #1 suggested to use the refund package as done by Goldsmith et al., as the choice of basis functions is not based on validation studies, and might therefore not be the optimal choice of model selection. We followed this suggestion and performed the respective analyses. Results were very similar. Please find a comparison between the FoSR with Daubechies Wavelets as basis functions and cubic b-splines as basis functions in figure 1 and figure 2, respectively. Per default, the refund package performs FoSR with cubic b-splines and 15 basis functions. Fifteen basis functions are similar to the choice of 18 basis functions made in our approach. Of note, the authors of the refund package do not ground their choice of basis functions or their numbers on statistical considerations.

Figure 1: Time course of absolute activity in activity counts between MCI and HOV. FoSR is modelled with 18 Daubechies wavelet basis functions.

Figure 2: Time course of absolute activity in activity counts between MCI and HOV. FoSR is modelled with 18 cubic B-spline basis functions based on the refund package.

Although we do acknowledge that to some extent more research could improve Daubechies Wavelets in use of FoSR, we believe that several critical points raised by Reviewer #1 are equally true for cubic splines. E.g., the choice of basis functions used is to some extent arbitrary. Additionally, it is assumed that activity patterns follow locally a polynomial progression of degree 3. Daubechies wavelets are 24 h periodic discrete functions and thus more suitable to model activity data that follows a 24h rhythm. The coefficient functions from FoSR analysis with cubic splines are 24 h periodic only because non-periodicity is averaged out in large samples.

We agree that the comparison FoSR with wavelets versus FoSR with cubic splines might be of interest for the informed reader so we added respective comparison in the supplementary appendix.

Comment #4:

The number of basis functions used in your wavelet transformation also dictates your capability to determine statistical significance at different times based on your use of a bonferonni correction to account for multiple comparisons. Why not use a false discovery rate correction instead? Applying Bonferroni based on number of basis functions seems to make it either too conservative or too liberal and has a direct impact on your capability to determine when the activity patterns differ between groups. Also, the refund package uses an FOSR function that is Bayesian and therefore utilizes confidence intervals to make determinations of significance rather than p-values.

The choice of correction for multiple testing is indeed an area in which more research might enhance the informative value of the FoSR. Of note, currently no such correction is implemented in the refund package. We here improved the model from Goldsmith et al to some extent. Bayesian methods also do not solve the problems arising from high dimensionality and bounding the FDR instead of the FWER does not change the fact that any method to adjust 288 highly correlated p-values tends to be too conservative. We compared our FoSR with correction for multiple comparison after Bonferroni-Holm (FWER) to one with Benjamini-Yekutieli (FDR). Both correction methods displayed the same time window with statistically significant differences in activity but FDR extended the time frame by ten minutes. Both methods are rather conservative but led to similar results. To spread awareness for the need to further validate different model choice in FoSR modelling we now added a sentence to the Discussion which now reads as follows:

Forth, model choices in FoSR such as number of basis functions or correction methods for multiple testing should be validated in different patient populations to enhance the informative value of the regression method.

Comment #5:

The authors also never run a multiple linear regression on the aggregate activity data. In the introduction the authors state that this method is in effectual but they never perform it to actually demonstrate that this is the case.

A comparison with well-known multiple regression assists the reader in putting the new approach into perspective. We thank reviewer #1 for this suggestion. The reason why we did not present this analysis in our manuscript was that a multiple linear regression on aggregated activity data does not take into account differential activity pattern over the time course of the day which we believe to be important. Therefore, we think that this comparison is actually misleading as it might suggest that the FoSR is a better regression model than a multiple linear regression. But the comparison is not correct in the first place as the linear model treats activity data as an absolute number and the FoSR treats it as a function of time-based activity values. We performed a linear mixed regression adjusted for age and sex on the average activity counts per day as a dependent variable and group as a covariate. The respective linear regression model was now added to the Methods and the Results section which now read as follows:

We performed a multiple linear regression with average activity count as a dependent variable and group as an explanatory variable adjusted for age and sex.

Based on the simple linear regression model, average activity count was similar in both groups (-7 counts for MCI compared to HOV, 95% CI -20 to 7, p-value = 0.32). 

Comment #6:

Change hours to hour, line 110.

We did the requested change.

Comment #7:

Be sure to cite the NHANES cohort, line 174.

We cited the NHANES 2003-2004 cohort accordingly:

Centers for Disease Control and Prevention National Center for Health Statistics. NHANES Questionnaires, Datasets, and Related Documentation [Internet]. [accessed 2021 Feb 20]. Available from: https://wwwn.cdc.gov/nchs/nhanes/continuousnhanes/default.aspx?BeginYear=2003

Comment #8:

Please cite who is suggesting that the use of cubic splines or wavelet transformations is suggested for actigraph data, line 206.

We added the requested references:

Xiao L, Zipunnikov V, Ruppert D, Crainiceanu C. Fast covariance estimation for high-dimensional functional data. Stat Comput. 2016;26(1–2):409–21.

Leise TL. Analysis of Nonstationary Time Series for Biological Rhythms Research. J Biol Rhythms. 2017;32(3):187–94.

---

## [Decision Letter · Decision Letter 2]

29 Apr 2021

Applying time series analyses on continuous accelerometry data – a clinical example in older adults with and without cognitive impairment

PONE-D-20-08099R2

Dear Dr. Rackoll,

We’re pleased to inform you that your manuscript has been judged scientifically suitable for publication and will be formally accepted for publication once it meets all outstanding technical requirements.

Kind regards,

Henrik Oster, Ph.D.

Academic Editor

PLOS ONE

Additional Editor Comments (optional):

Reviewers' comments:

Reviewer's Responses to Questions

**Comments to the Author**

1. If the authors have adequately addressed your comments raised in a previous round of review and you feel that this manuscript is now acceptable for publication, you may indicate that here to bypass the “Comments to the Author” section, enter your conflict of interest statement in the “Confidential to Editor” section, and submit your "Accept" recommendation.

Reviewer #1: All comments have been addressed

2. Is the manuscript technically sound, and do the data support the conclusions?

Reviewer #1: Yes

3. Has the statistical analysis been performed appropriately and rigorously? 

Reviewer #1: Yes

4. Have the authors made all data underlying the findings in their manuscript fully available?

Reviewer #1: Yes

5. Is the manuscript presented in an intelligible fashion and written in standard English?

Reviewer #1: Yes

6. Review Comments to the Author

Reviewer #1: I commend the authors on their dedication to this manuscript! I wish them the best of luck on their continued research.

7. PLOS authors have the option to publish the peer review history of their article (what does this mean?). If published, this will include your full peer review and any attached files.

Reviewer #1: **Yes: **Andrew Hooyman

---

## [Editor Report · Acceptance letter]

4 May 2021

PONE-D-20-08099R2 

Applying time series analyses on continuous accelerometry data – a clinical example in older adults with and without cognitive impairment 

Dear Dr. Rackoll:

I'm pleased to inform you that your manuscript has been deemed suitable for publication in PLOS ONE. Congratulations! Your manuscript is now with our production department. 

Kind regards, 

on behalf of

Prof. Henrik Oster 

Academic Editor

PLOS ONE